# Direct experimental observation of blue-light-induced conformational change and intermolecular interactions of cryptochrome

Pei Li[1,7], Huaqiang Cheng [1,7], Vikash Kumar[2], Cecylia Severin Lupala [2], Xuanxuan Li[2], Yingchen Shi [2], Chongjun Ma[1], Keehyoung Joo [3], Jooyoung Lee[4], Haiguang Liu [2,5,6✉] & Yan-Wen Tan [1✉]

Cryptochromes are blue light receptors that mediate circadian rhythm and magnetic sensing in various organisms. A typical cryptochrome consists of a conserved photolyase homology region domain and a varying carboxyl-terminal extension across species. The structure of the flexible carboxyl-terminal extension and how carboxyl-terminal extension participates in cryptochrome's signaling function remain mostly unknown. In this study, we uncover the potential missing link between carboxyl-terminal extension conformational changes and downstream signaling functions. Specifically, we discover that the blue-light induced opening of carboxyl-terminal extension in *C. reinhardtii* animal-like cryptochrome can structurally facilitate its interaction with Rhythm Of Chloroplast 15, a circadian-clock-related protein. Our finding is made possible by two technical advances. Using single-molecule Förster resonance energy transfer technique, we directly observe the displacement of carboxyl-terminal extension by about 15 Å upon blue light excitation. Combining structure prediction and solution X-ray scattering methods, we propose plausible structures of full-length crypto-chrome under dark and lit conditions. The structures provide molecular basis for light active conformational changes of cryptochrome and downstream regulatory functions.

[1] State Key Laboratory of Surface Physics, Department of Physics, Fudan University, 200433 Shanghai, China. [2] Beijing Computational Science Research Center, 100193 Beijing, China. [3] Center for Advanced Computation, Korea Institute for Advanced Study, Seoul 02455, Republic of Korea. [4] School of Computational Sciences, Korea Institute for Advanced Study, Seoul 02455, Republic of Korea. [5] Physics Department, Beijing Normal University, Haidian, 100875 Beijing, People's Republic of China. [6]Present address: Microsoft Research AI4Science, Beijing, China. [7]These authors contributed equally: Pei Li, Huaqiang Cheng. ✉email: hgliu@csrc.ac.cn; ywtan@fudan.edu.cn

Cryptochromes (CRYs) are blue-light receptors with multiple signaling functions in plants and animals. The most well-known function of CRYs is the entrainment of the circadian rhythm. In plants, CRYs regulate gene expression to control the inhibition of hypocotyl elongation and the initiation of flowering[1]. CRYs also entrain the circadian rhythm in insects and act as the primary repressor of the central oscillator in mammals[2–4]. CRYs have also been found to participate in the magnetic navigation of insects and migratory birds[5–8].

Structurally, CRY photolyase family (CPF) proteins possess a relatively conserved photolyase homology region (PHR) domain and a vastly varying carboxyl-terminal extension (CTE), which can comprise tens to hundreds of amino acids. The PHR domain harbors a noncovalently bound cofactor flavin adenine dinucleotide (FAD), which contributes to the blue-light-sensing ability of CRYs[9,10]. Although CTEs vary across species in both length and an amino acid sequence, they have been discovered to be a key element of CRY-dependent signal transduction[11]. For instance, CTE-overexpressed *Arabidopsis* seedlings exhibit the short-hypocotyl phenotype upon blue-light illumination, indicating that the CTE of the *Arabidopsis* CRY mediates a constitutive light response[12]. The exon 11 in the CTE of human CRY1 deletion mutant lengthens the circadian period to cause delayed sleep phase disorder (DSPD), suggesting that the CTE is necessary and sufficient to control circadian timing by regulating its association with CLOCK:BMAL1[13]. The presence of CTE affects the quantum yield (QY) of radical pair and the order-disorder transition of the phosphate binding motif in *Columba livia* CRY4 PHR domain (PBM motif, 235–245 aa)[14]. Deletion of residues 521–540 from the CTE of the *Drosophila melanogaster* CRY decreases the neuronal sensitivity to magnetic fields[15], which implies that this region is necessary for magnetic sensing. Despite the importance of CTEs for various functions of CRYs, the molecular mechanism through which CTEs realize these functions remain unclear, mainly due to the lack of structural information on CTEs. Nonetheless, several models have been proposed for CTE functions. The CTEs in plant CRYs have been shown to be targets for extensive blue-light-dependent phosphorylation and to be electrostatically repelled from the PHR domain to adapt open conformations, resulting in exposure of the NC80 motif (an 80-residue region located between the N-terminal PHR domain and the CTE of *Arabidopsis* CRY2) and signal transduction that triggers photomorphogenic responses[16,17]. In *D. melanogaster*, the light-induced interactions between *Dm*CRY, Timeless, and the E3 ubiquitin ligase Jetlag are thought to rely on the conformational changes of the *Dm*CRY CTE[18,19]. Nonetheless, direct measurements of residue distances are challenging, especially at the single-molecule level. In a recent study, Chandrasekaran et al. applied 4-pulse double electron-electron resonance (4P-DEER) method to investigate light-induced enhanced motion of CTE in *Dm*CRY[20]. However, the structural information obtained by these bulk methods is insufficient to build atomic resolution structures. And the distance constraints necessary to refine the CTE structure are still lacking for *Cra*CRY. Furthermore, the CTEs are presumably flexible and, therefore, difficult to crystallize[21] and resolve to atomic resolution. Currently, among all experimentally determined CRY structures, only *Dm*CRY has been resolved to its full length (Protein data bank [PDB] code: 4GU5/3TVS)[22,23]. However, the CTE of *Dm*CRY is composed of only 20 residues, whereas some CTEs comprise more than 100 residues. Therefore, obtaining structural information on full-length CRYs is a challenging task, but highly desirable because it would enable the examination of light-induced conformational changes and yield a better understanding of CTE-related functions.

According to their evolutionary sequence relationships, the CRY family is categorized into three types: the animal type, the plant type, and CRY-DASH (DASH: *Drosophila, Arabidopsis, Synechocystis, Human*). Animal-type CRYs comprise four sub-branches: animal CRY1, animal CRY2, animal-like CRY, and insect CRY. Intriguingly, on the evolution tree, some algal CRYs have been sorted between animal CRYs and insect CRYs and have thus been named animal-like CRYs (aCRYs)[24,25]; aCRY orthologs from *Ostreococcus tauri* (*Ot*CPF1)[26] and *Phaeodactylum tricornutum* (*Pt*CPF1)[27] have been discovered to have bifunctional activities, acting as both a transcriptional regulator of the circadian clock and a (6-4) photolyase for DNA repair.

*Chlamydomonas reinhardtii* (*C. reinhardtii*) is a model system for photosynthesis and the molecular basis of the circadian clock[28,29]. *C. reinhardtii* has one animal-like ortholog (*Cra*CRY), one plant CRY, and two CRY-DASHs. Among these, *Cra*CRY both regulates circadian gene transcription and repairs ultraviolet-DNA lesions in a blue-light-dependent manner[21,30]. Unlike photolyases, *Cra*CRY harbors an elongated CTE domain (A497–E595 a.a.), making it a favorable model for investigating light-induced conformational changes in CRY and the dynamics of CTEs.

To understand CRY signaling functions, the molecules that directly interact with *Cra*CRY for downstream signaling need to be identified. The clock components of *C. reinhardtii* include RNA-binding protein complex CHLAMY1[31], a casein kinase[32], and Rhythm Of Chloroplast (ROC) proteins[33]. While the putative transcription factor ROC15 should be one of the most likely interaction partners to *Cra*CRY because of its light-dependent changes in expression level[34]. In this study, we discovered blue-light-induced intermolecular interaction between *Cra*CRY and ROC15 by conducting in vitro pull-down assays[35], single-molecule assays, and in vivo bimolecular fluorescent complementation (BiFC)[36].

Relevant to CRYs interactions with other signaling proteins, dimerization of CRYs was thought to play a role in their functions. Nonetheless, it remains controversial whether the physiological active states of CRY induced by blue-light are in dimer or oligomer forms[37,38]. In *Arabidopsis* CRY-CIB1 (CRY-interacting basic-helix-loop-helix 1) modulation of transcription[37], the binding affinity between CIB1 and the stable *At*CRY2 dimer mutant was higher than that for the wild-type *At*CRY2[38]. This implied the *At*CRY2 dimer was physiologically active. By contrast, in a CRY-BIC (blue-light inhibitor of CRY) negative-feedback system, the BIC binds to monomeric *At*CRY2 to suppress the blue-light-dependent dimerization and physiological activities[39]. Moreover, cryogenic electron microscopy (Cryo-EM) and X-ray crystal diffraction studies on the structures of the blue-light-activated *At*CRY2 (PHR domain) and the stable *At*CRY2 dimer mutant (PHR domain) have revealed stable tetramers[40–42]. Clearly, whether the active state of CRYs is dimeric or monomeric is a critical but unanswered question.

To gain a thorough understanding of the molecular mechanisms of *Cra*CRY, we applied single-molecule techniques combined with structure prediction, solution small-angle X-ray scattering (SAXS) experiments, and molecular dynamics (MD) simulations to investigate the functional dynamics of *Cra*CRY. Single-molecule Förster resonance energy transfer (smFRET)[43] data revealed that upon blue-light exposure, the CTE was partially dissociated from the PHR domain, reflected by a distance increase of approximately 15 Å, between the C317 (in the PHR domain) and E595 (in the CTE) residues. Full-length structures of *Cra*CRY, predicted using homology modeling, were refined to fit the SAXS data obtained in both dark and lit conditions by using SAXS-driven MD simulations. Refined structures, supplemented by protein-docking analysis, provided molecular details regarding light-induced conformational changes, *Cra*CRY dimerization, and *Cra*CRY–ROC15 complex formation. Furthermore, our

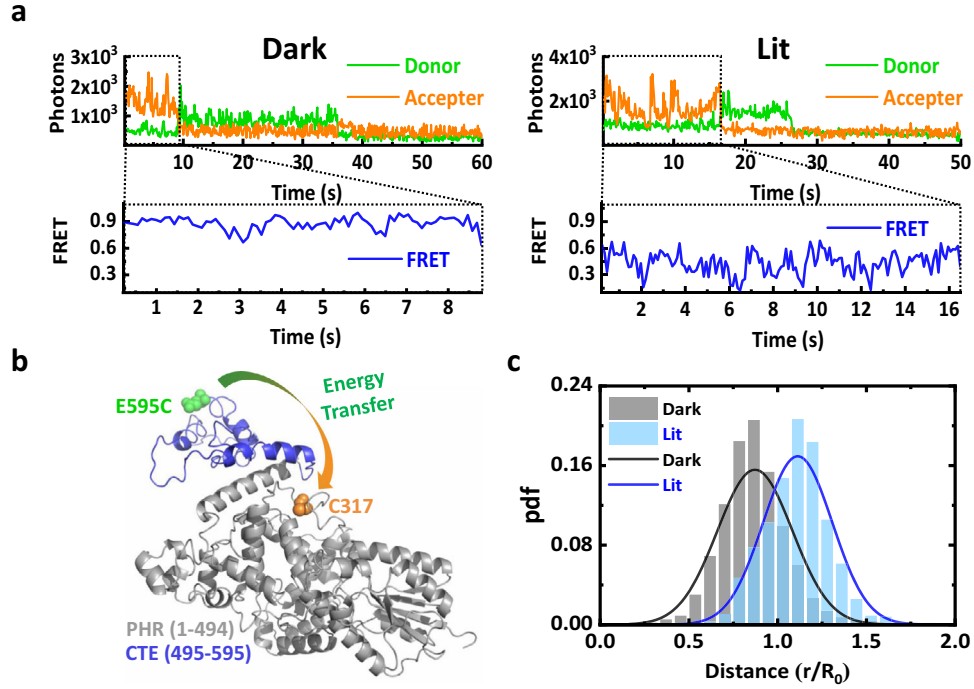

**Fig. 1 Blue-light-induced conformational change of *Cra*CRY determined using smFRET. a** Single-molecule fluorescence time traces show the donor and acceptor fluorescence intensity with donor excitation under dark or lit conditions (upper panels). Single-molecule FRET time traces show the relative FRET efficiency (lower panels). **b** *Cra*CRY is labeled with Atto550 (green)/Atto647 (orange) as the donor and acceptor at E595C or C317 sites, respectively. The PHR and CTE domains are shown in gray and blue, respectively. The predicted structure of *Cra*CRY is shown to illustrate the labeled sites. **c** smFRET data converted into distance histograms of *Cra*CRY molecules under dark or lit conditions. The normalized probability distribution functions (pdfs, shown in bars) are fitted with Gaussian distributions (solid lines).

single-molecule assays revealed that the blue-light-induced *Cra*CRY–ROC15 interaction and *Cra*CRY dimerization were mutually exclusive. The results from this study refine the current understanding and provide a light activation model for CRYs.

## Results
**Blue-light-induced conformational change of *Cra*CRY probed by smFRET**. For smFRET experiments, we mutated five cysteines of wild-type *Cra*CRY (Supplementary Table S1 and Supplementary Fig. S1). The native C317 and a mutated residue at the C-terminal E595C were used as labeling residues (Fig. 1, source data in Supplementary Data 1 and 2). The smFRET signals were measured in both dark and lit conditions; typical time traces are shown in Fig. 1a. The distribution of the distance between acceptor and donor residues (C317 and E595C, see Fig. 1b, source data in Supplementary Data 1 and 2) is displayed in Fig. 1c. Fitting the distance distributions with Gaussian functions revealed that the distance between C317 and E595C was $50 \pm 13$ Å for the dark state, while, the distance extended to $65 \pm 12$ Å upon blue-light illumination (i.e., the lit state; Supplementary Table S2). The histograms obtained under dark and lit conditions include 2496 data points from 16 molecules and 2869 data points from 22 molecules, respectively. Certainly, shorter distances revealed by smFRET for dark state *Cra*CRY suggest that the conformation is more compact than the lit state, which exhibits larger distances. Considering that the residues 317 and 595 are located in PHR and CTE, respectively, the increased distance suggests blue-light-induced conformational changes that CTE moves away from PHR. The conformational changes of CRYs directly observed at the single-molecule level are important for the understanding of CRY signaling mechanism.

To eliminate potential interference with the smFRET signals from the dimerization of *Cra*CRY, monomeric *Cra*CRY was immobilized on the slide by using bivalent streptavidin with biotin-photocleavable(PC)-DNA[44], which ensured a unique immobilization site on each streptavidin molecule (Supplementary Fig. S2a, b). This immobilization scheme ensured monomeric tethering and prolonged observation of the same molecule. Additionally, the monomeric tethering prevented fluorescent interference from dimerization. To assess the monomeric tethering efficiency, we designed control experiments under dark conditions using three monomeric fluorescent proteins as probes. The fluorescent proteins mCitrine, mOrange2, and mCherry were fused with *Cra*CRY, and we evaluated the percentage of trajectories exhibiting single- or double-step bleaching. In this single-molecule assay, most of the fused proteins underwent only one bleaching step, corresponding to the monomeric form of *Cra*CRY. The fractions of double-step bleaching for *Cra*CRY::mCitrine, *Cra*CRY::mOrange2, and *Cra*CRY::mCherry were 6.8%, 8.6%, and 7.8%, respectively (Supplementary Fig. S2c). A similar level of double-step bleaching trajectories was observed in our single-molecule experiments, and thus, we attributed this ~8% to the background of our measurement. The double-step bleaching was attributed to the following factors: molecules embedded in the same diffraction-limited fluorescent spot, impurities, defective biotin-PC-DNA, or intrinsic dimeric forms of *Cra*CRY/fluorescent proteins. The control experiments performed on the immobilized molecules demonstrated that the *Cra*CRY remained as monomers under dark conditions, guaranteeing that the smFRET conformational data of monomeric *Cra*CRY under lit conditions would not be affected by blue-light-induced dimerization.

**X-ray scattering and MD simulations revealed structural differences induced by blue-light illumination**. Structural information on CTE domains obtained through X-ray crystallography

or other high-resolution structural methods is limited. To probe the states and structures of *Cra*CRY in solution, SAXS signals of full-length *Cra*CRY or its PHR domain were measured under both dark and lit conditions at the BioSAXS beamline of the Shanghai Synchrotron Radiation Facility (SSRF) (Fig. 2a, source data in Supplementary Data 2). Inline size-exclusion chromatography (SEC) was applied during SAXS measurement to separate monomeric *Cra*CRY from its dimeric forms (Supplementary Fig. S3, S4). The dimensionless Kratky plots indicate that the PHR domain has a compact conformation with a signature peak that matches well-folded protein structures (Fig. 2b, source data in Supplementary Data 2). The crystal structure of the *Cra*CRY PHR domain (PDB ID: 5ZM0) is in good agreement with the SAXS data and the reconstructed low-resolution model (Fig. 2d, e). For the full-length *Cra*CRY, the peak positions are shifted to the right in the dimensionless Kratky plots, suggesting flexibility of the CTE or the elongated shape of full-length *Cra*CRY. The structure of the full-length *Cra*CRY was predicted using SAXS data-assisted method and then further refined against SAXS data using SAXS-driven MD simulations for both dark and lit states (see Methods for details). The refined structures quickly converged to two conformations, whose theoretical profiles fit the SAXS data for dark and lit states. Although it is difficult to assign the two structure ensembles to dark or lit states solely based on their fitting to SAXS data (Supplementary Fig. S5), the distance information obtained from smFRET experiments can be applied to assign the two structure ensembles to dark and lit states (Fig. 2c). Two representative models were superposed to the respective low-resolution envelopes constructed from SAXS profiles (Fig. 2d, source data in Supplementary Data 1). According to refined structures, the CTE of *Cra*CRY is partially packed onto the PHR domain in the dark state, while the CTE is in a more tilted orientation. The difference in CTE orientations results in a lateral shift of about 22 Å in the lit state relative to its dark state position (Fig. 2d, f). The tilt of CTE in the lit state is consistent with the pairwise distance distribution analysis, which reveals a more extended conformation of *Cra*CRY in the lit state than in the dark state (Fig. 2g and Supplementary Fig. S6, source data in Supplementary Data 2). The distances between C317 and E595 are ~45 Å and ~62 Å in the representative structures for dark and lit states, respectively (measured between the Cα atoms of two residues, see Fig. 2d; Supplementary Movies S1, S2). The details on distances measured from two structure ensembles are summarized in the supplementary information (Supplementary Fig. S5c), showing a consistent trend with the dynamical features observed in smFRET experiments (see Fig. 1c).

**Light-induced intermolecular interactions discovered by pull-down, single-molecule, and BiFC assays in vitro and in vivo.** Although *Cra*CRY has been identified as bifunctional at the cellular level[21], its downstream signaling partners remain unclear. Based on the function of *Cra*CRY in entraining the circadian rhythm[28,31–34], we investigated the interaction between *Cra*CRY and ROC15. Both full-length ROC15 and the glutamic acid-rich protein domain ROC15(GARP)[33] (M377–M445 a.a.) constructs were analyzed in experiments.

For in vitro glutathione S-transferase (GST) pull-down assays[35], the purified His-tag-fused *Cra*CRY, and GST-fused ROC15(GARP) proteins were incubated together at room temperature in the dark for 20 min (Fig. 3a). Half of the mixture was then injected into a GST affinity column directly under the dark condition. The other half was illuminated with blue light (450 nm LED lamp) for 5 min and then subjected to the same operation as the first half of the mixture. The eluents of different constructs from the GST affinity column are shown in Fig. 3b.

Sodium dodecyl sulfate–polyacrylamide gel electrophoresis (SDS-PAGE) revealed that the full-length *Cra*CRY could be baited by GST-fused ROC15(GARP) proteins only upon blue-light illumination (Fig. 3b, left and Supplementary Fig. S7, S8). The GST pull-down assay for the PHR domain of *Cra*CRY (M1–A489 a.a.) further revealed that the PHR was the structure domain of interaction between *Cra*CRY and ROC15(GARP) (Fig. 3b, middle). No interaction was detected between the CTE domain (V463-E595 a.a.) and ROC15(GARP) under either the dark or lit condition (Fig. 3b, right).

To determine whether the interaction between ROC15 and *Cra*CRY occurs in vivo, we conducted BiFC assays in *C. reinhardtii* cells. Full-length ROC15 labeled with split mCitrine C-terminal was cotransfected with N-terminal mCitrine-labeled *Cra*CRY into live *C. reinhardtii* (Fig. 3f). The difficulty of performing a BiFC assay in live algal cells is that upon blue-light illumination, strong autofluorescence is generated (Fig. 3g)[45,46]. Additionally, the expression of *Cra*CRY and ROC15 proteins may alter the circadian rhythm; therefore, we had to track their interaction for at least 24 h (Supplementary Fig. S9). To discern the true BiFC signal of the recombined mCitrine from the autofluorescent background, we performed fluorescence-lifetime imaging. The fluorescence lifetime of mCitrine was 3.09 ± 0.04 ns at 25 °C, close to the 2.92 ± 0.07 ns at 37 °C reported in the literature[47] and substantially longer than that for the native pigments of plants. In the BiFC experiments[36], the fluorescence lifetime of ~2.9 ns observed only in doubly transfected cells upon blue-light illumination indicated that the two halves of mCitrine were recombined by the interaction between *Cra*CRY and full-length ROC15 (Fig. 3g, source data in Supplementary Data 2).

To obtain the quantitative fraction and stoichiometry of the *Cra*CRY–ROC15 interaction, we conducted single-molecule pull-down experiments. The *Cra*CRY fused with mCitrine (*Cra*CRY::mCitrine), and the ROC15(GARP) fused with mCherry (ROC15(GARP)::mCherry) were used in this experiment. The single-molecule pull-down results (Fig. 3c, d) were consistent with the GST pull-down findings in vitro, demonstrating that the interaction between *Cra*CRY and ROC15 was blue-light-dependent. The colocalization signals indicated the fraction of interacting *Cra*CRY was 23.3 ± 8.8% (Fig. 3e, Supplementary Table S3). The data consistently demonstrated that ROC15 was a direct interaction partner of *Cra*CRY for blue-light-dependent downstream signaling.

**CraCRY–ROC15 interaction and CraCRY dimerization are blue-light-dependent and mutually exclusive.** Light-induced dimerization and oligomerization of *Cra*CRY have been demonstrated using SEC assays[48,49], but the details are controversial. The SEC results obtained by Franz-Badur et al. revealed that most *Cra*CRY (93–96%) preferred to remain as monomers under dark conditions, whereas blue-light illumination (at 450 nm) induced partial dimerization of *Cra*CRYs[49]. However, Oldemeyer et al. found that red-light illumination (at 633 nm) promoted the transformation of dimeric *Cra*CRYs into oligomers, yet control experiments on blue-light-induced behaviors of *Cra*CRY were lacking in that study[48]. Spexard et al. investigated the light-induced changes using Fourier transform infrared (FTIR) difference spectroscopy method and reported flavin oxidization due to blue light without detectable changes in secondary structures[49]. In the present study, to reveal the blue-light-dependent dimerization or oligomerization behavior of *Cra*CRYs, both SEC and single-molecule photobleaching screening assays were performed. The SEC results demonstrated that *Cra*CRYs transformed from monomers into dimers upon blue-light illumination, which is consistent with the observation

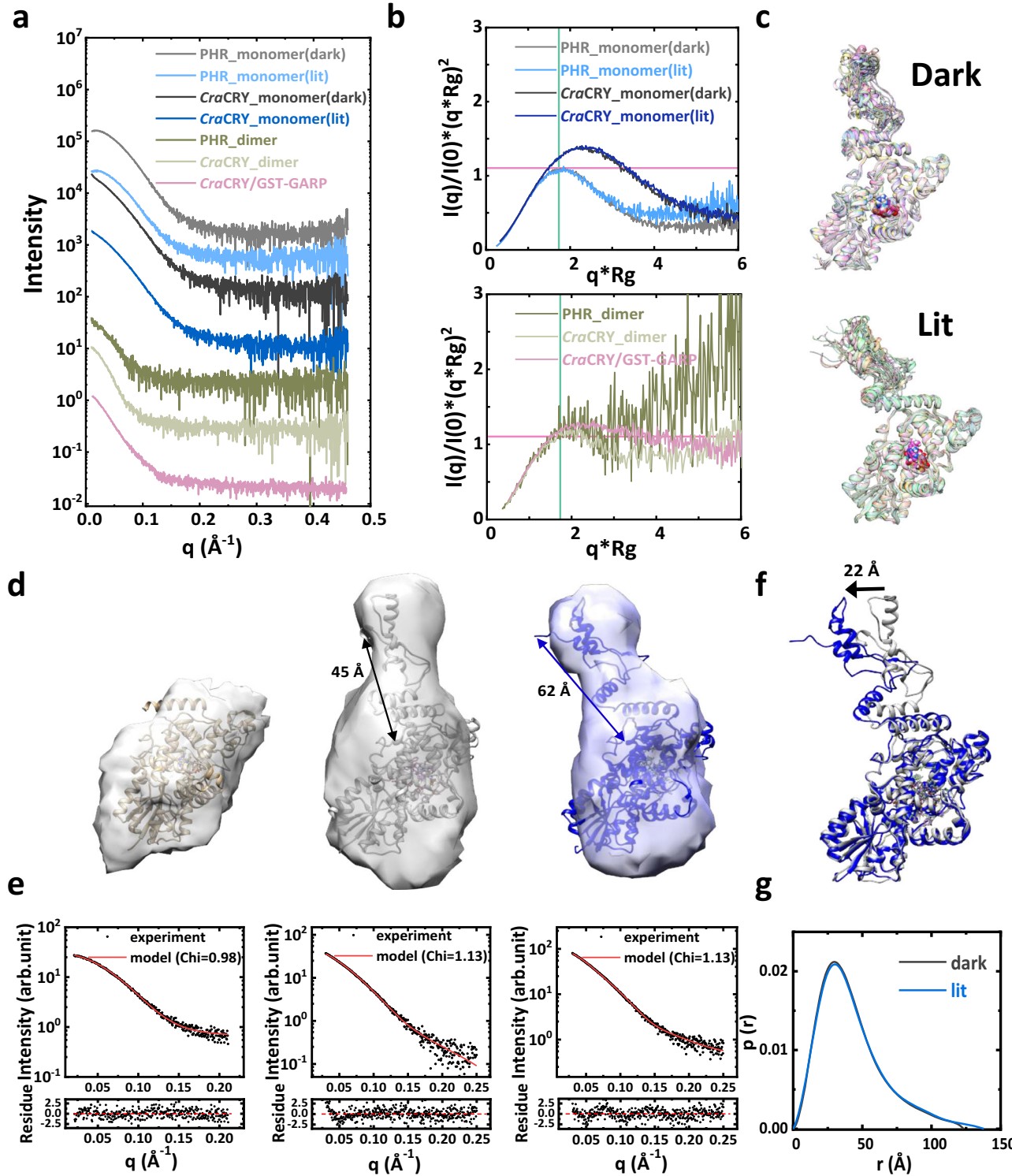

**Fig. 2 SAXS data and modeling. a** SAXS data in the range of measurement. The intensities were scaled for presentation purposes. **b** Dimensionless Kratky plots for the SAXS data. **c** Structure ensembles obtained from SAXS-driven MD simulations. **d** Reconstructed low-resolution models superposed to the crystal structure of PHR and the representative models of *Cra*CRY in dark and lit states, shown in panels from left to right, respectively. **e** The theoretical SAXS profiles (solid red) calculated from the atomic models, fitted to experimental data (black dots). The residual plots ($I_{expt} - I_{model}$)/$\sigma_{expt}$ are shown underneath each fitting result. The figures are arranged left to right showing the SAXS data of PHR, *Cra*CRY in dark, and *Cra*CRY in lit, respectively. **f** The positions of CTE in representative structures, with dark state structure shown in gray color, and the lit state in blue (see panel **d**). **g** The pairwise distance distribution function p(r) derived from SAXS data in dark and lit states.

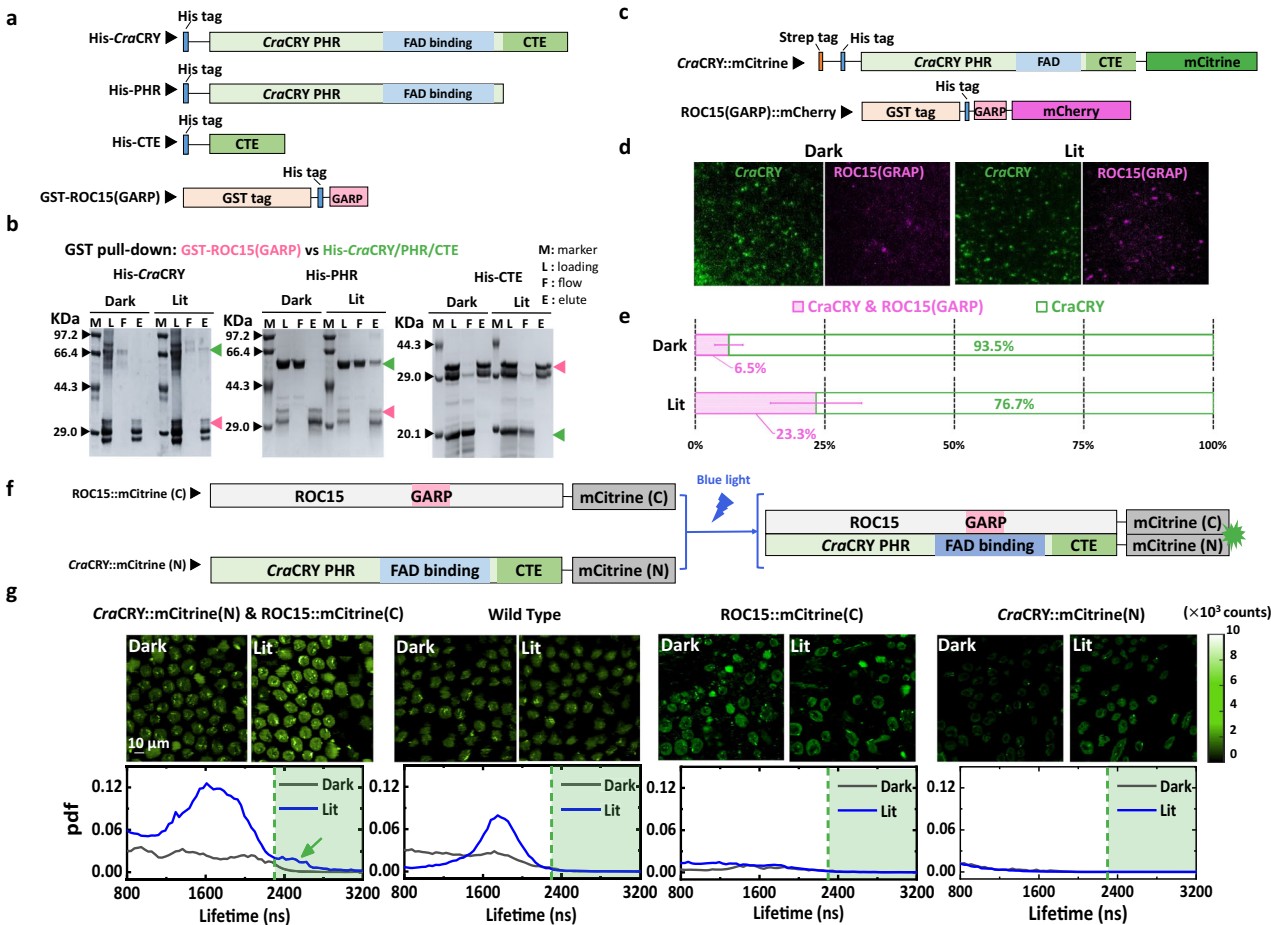

**Fig. 3 Intermolecular interactions identified using pull-down and BiFC assays. a** Schematic of protein construct design for the GST pull-down assays. His-tags were fused with full-length *Cra*CRY, the PHR domain, and the CTE domain. They were preyed on by GST-tag fused ROC15(GARP) as the bait protein. **b** Pull-down molecular complex analyzed using SDS-PAGE. The green arrows mark the band of wild-type *Cra*CRY (left), the PHR domain (middle), and the CTE domain (right). The pink arrows mark the band of ROC15(GARP). **c** Protein construct design for single-molecule pull-down assays. Fluorescence proteins mCitrine and mCherry are fused with full-length *Cra*CRY and ROC15(GARP), respectively. As the bait protein, Strep-tag-fused *Cra*CRY::mCitrine is tethered to the glass slide by streptavidin. **d** Total internal reflection fluorescence microscopy (TirfM) images of single-molecule pull-down assays. Green and magenta points are the fluorescent signals emitted from *Cra*CRY::mCitrine and ROC15(GARP)::mCherry, respectively. **e** Statistics of the colocalized molecules in single-molecule pull-down assays. The colocalization fractions of *Cra*CRY::mCitrine and ROC15(GARP)::mCherry complex under dark and lit conditions are shown as magenta bars, whereas green bars indicate the fractions of protein complex containing only *Cra*CRY. The error bars are the statistical standard deviation, seeing Supplementary Table S3 for exact values. **f** Sketch map of protein designs and principles for BiFC assays. Upon blue-light illumination, the binding between *Cra*CRY and ROC15 triggered the recombination of mCitrine(N) and mCitrine(C), and the fluorescence signal of mCitrine was detected using fluorescence-lifetime imaging microscopy (FLIM). **g** Fluorescent images (upper) and lifetime distributions (lower) of single- and double-transfected and wild-type *C. reinhardtii* cells under the dark or lit condition. Fluorescent intensity images are shown in green. Gray and blue lines indicate the lifetime distributions for each image under the dark and lit conditions, respectively. The green arrow points to the characteristic lifetime component of mCitrine signals.

by Franz-Badur et al. (Supplementary Fig. S3a). In single-molecule photobleaching screening assays, by counting the photobleaching steps of *Cra*CRY::mCitrine protein complexes under dark and lit conditions, the population of two-step bleaching increased from 7.5 ± 2.7% to 25.1 ± 5.9% after exposure to blue light (Fig. 4a–c, Supplementary Table S4, source data in Supplementary Data 2). These percentages of two-step trajectories in the dark and lit experiments also matched the SEC results observed by ourselves and Franz-Badur et al.[50]. To quantitatively analyze *Cra*CRY dimer formation, we designed and executed another smFRET experiment. In this smFRET assay, 65 dimers (22,591 data points) were identified. The intramolecular distance between C317 and E595C was 48 ± 12 Å, more consistent with the dark conformation of *Cra*CRY (50 ± 13 Å) over the lit conformation (Fig. 4d, e and Supplementary Table S2, source data in Supplementary Data 1 and 2).

The dimer structure of the *Cra*CRY PHR domain was constructed based on the structure of a plant CRY recently resolved using Cryo-EM single-particle imaging method[38]. After superimposing the crystal structure of the PHR domain (PDB code: 5ZM0) on the Cryo-EM model, we obtained a dimer structure model for PHR domain, of which the theoretical SAXS profile is in good agreement with the experimental data (Supplementary Fig. S10a, the third model, named as the head-to-head model hereafter). Besides the constructed dimer model based on the Cryo-EM structure, we also generated alternative dimer models based on crystal packing or biological assembly structures (see Supplementary Fig. S10a for two alternatives). As shown in Figure S10a, all three PHR dimer models agree with SAXS data, yielding similar chi-scores between theoretical profiles and experimental data. We selected the most plausible dimer model based on two additional criteria: (1) the head-to-head model is consistent with the low-

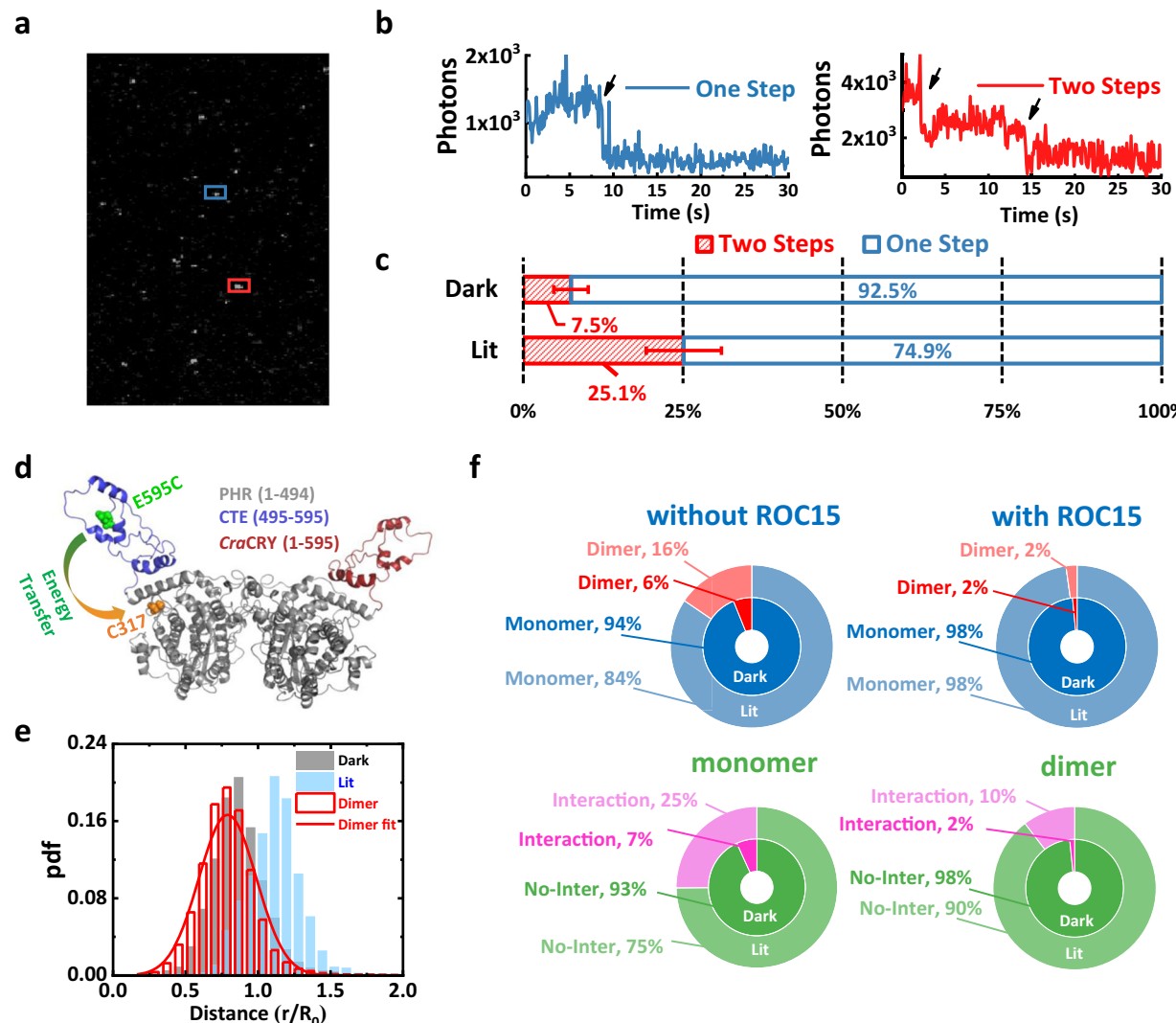

**Fig. 4 Blue-light-induced dimerization and *Cra*CRY–ROC15 interaction are mutually exclusive. a** Fluorescence TirfM image of *Cra*CRY::mCitrine proteins upon blue-light illumination. Every bright spot corresponds to an isolated protein complex. The point in the blue box denotes a protein complex, with one-step bleaching (blue trajectory in panel **b**) detected in this protein complex, whereas the protein complex in the red box exhibited two-step bleaching (red trajectory in panel **b**). **b** Single-molecule fluorescence trajectories showing the mCitrine fluorescence intensity with one (blue) or two (red) bleaching steps. **c** Monomer (blue) and dimer (red) fractions under dark and lit conditions. The error bars are the statistical standard deviation, see Supplementary Table S4 for exact values. **d** Schematic representation of the smFRET assays for *Cra*CRY dimer. The CTE, PHR, and labeling sites for smFRET assays are shown in the figure; the red colored cartoon shows the other *Cra*CRY molecule (no FRET labeling) in the dimer model. **e** Distribution of the distance between C317 and E595C of the *Cra*CRY in dimeric state, shown as a red histogram. The gray and blue histograms are the distance distributions of monomers in dark and lit states, respectively. The red solid line shows the fitting results to the distances in dimeric *Cra*CRY. **f** Mutually exclusive relationship between *Cra*CRY dimerization and *Cra*CRY–ROC15 intermolecular interaction. The blue and red donut charts show the dimer fractions (red) with or without ROC15 under the dark (dark blue and dark red) and lit (light blue and light red) conditions. The green and magenta donut charts show the *Cra*CRY–ROC15 interaction fractions with or without *Cra*CRY dimerization under the dark (dark green and dark magenta) and lit (light green and light magenta) conditions. The experiments demonstrated the mutually exclusive relationship between *Cra*CRY dimerization and *Cra*CRY–ROC15 intermolecular interaction.

resolution envelope model; (2) the head-to-head model does not introduce steric clashes when constructing the dimer structure of full-length *Cra*CRY, while the other two alternative models result in unphysical clashes at the CTE. Considering that the dimeric *Cra*CRY possesses conformations similar to its dark state as revealed in smFRET experiments, the dimer model for full-length *Cra*CRY was constructed by superposing the dark state structure to the Cryo-EM structure template. The resulting model is also in good agreement with the SAXS data measured in the SEC-SAXS experiment (Supplementary Fig. S10b).

Single-molecule interaction assays were performed to investigate the relationships between *Cra*CRY dimerization and

*Cra*CRY–ROC15 interaction. By combining bleaching step counting with colocalization analysis, we discovered a mutually exclusive relationship between *Cra*CRY dimerization and *Cra*CRY–ROC15 interaction (Fig. 4f). The populations of blue-light-induced dimers with and without ROC15 were measured. Without colocalized ROC15, the dimer fraction increased from $6.1 \pm 2.1\%$ (dark) to $15.5 \pm 5.1\%$ upon blue-light illumination, which was consistent with the *Cra*CRY::mCitrine-only measurement. By contrast, with colocalized ROC15, the dimer fraction of *Cra*CRY was only 1–2% under both the dark and lit conditions, which we attributed to the background level of our single-molecule experiment. The same dataset was analyzed for

differences in the interaction between ROC15 with either *Cra*CRY monomer or dimer. The results revealed that the percentage of interaction between ROC15 and *Cra*CRY monomers increased to 25.1 ± 8.8% upon blue-light illumination, but the percentage of ROC15 interaction with dimers was only 10.4 ± 11.8% (Fig. 4f, Supplementary Table S5), substantially weaker than the interaction between ROC15 and *Cra*CRY monomers. The data indicate that blue-light-induced dimerization and *Cra*CRY–ROC15 interaction are mutually exclusive.

To investigate the interaction interface between *Cra*CRY and ROC15, we applied SEC-SAXS experiments to the mixture of GST-ROC15(GARP motif only) and *Cra*CRY. Because the structure of the GARP motif has not been experimentally determined, we predicted its structures using the Raptor-X server[51]. The predicted structure of GARP motif reveals a rigid domain, highly similar to the structure of its homologues (such as the GARP family from *Arabidopsis thaliana*[52]), allowing rigid body docking onto *Cra*CRY to investigate the plausible complex structures. Therefore, we applied a molecular docking method, ZDOCK[53], to predict the structures of the *Cra*CRY–ROC15(GARP) complex. The GARP motif of ROC15 was docked to a location in the vicinity of the FAD binding site, according to the ten docked structures with the highest docking scores. The Raptor-X was also applied to predict the structure of GST-tagged GARP domain (GST-GARP), which was built into the best complex model of *Cra*CRY–ROC15(GARP) (Supplementary Fig. S10c). The *Cra*CRY–ROC15(GST-GARP) complex structure is consistent with the SAXS data measured in the SEC-SAXS experiments. Interestingly, the structures of *Cra*CRY dimer and the *Cra*CRY–ROC15(GARP) complex revealed clues about the exclusive relation between *Cra*CRY dimerization and *Cra*CRY–ROC15 interaction: the dimerization interface is partially overlapped with the *Cra*CRY–ROC15 complex interface (Supplementary Fig. S10d).

## Discussion

Deciphering the light-dependent molecular mechanism of light-sensing molecules is an intriguing task. In this study, we systematically investigated the dynamics of *Cra*CRY, including the conformational changes, dimer formation, and its interactions with ROC15. Because of the flexible CTE domain in *Cra*CRY, ensemble-based methods provide only limited structural and dynamics information. Therefore, we employed single-molecule techniques and integrated the data with information acquired using other approaches, including SEC-SAXS, homology modeling, MD simulations, and pull-down assays. Advanced single-molecule assays enabled us to quantitatively characterize the light-dependent dynamics and investigate the underlying relationships.

CTE conformational changes and rearrangement relative to the PHR domain have long been speculated to be crucial to CRY functions that are activated by blue light. However, obtaining the structure of full-length CRYs has been difficult due to the CTE domain being flexible or disordered. Therefore, only indirect evidence has been obtained on CTE conformational changes to substantiate the speculation. For example, the conformational changes of CTE have been observed using in vitro trypsin proteolysis and time-resolved scattering experiments for *Dm*CRY[54,55] or in vivo double immunofluorescence labeling for chicken CRY1a[56]. In this paper, we present the direct measurement of the structural dynamics of full-length *Cra*CRY by probing light-induced conformational changes using smFRET, from which we quantitatively determined a ~15 Å extension of *Cra*CRY CTE from its PHR (Fig. 1c) from the dark to the lit state. Combined with the results of SEC-SAXS experiments and structure modeling, the full-length structures of *Cra*CRY under

dark and lit conditions (Fig. 2c) were constructed. To assess the potential model bias originating from the prediction method, we applied trRosetta server[57] to predict full-length *Cra*CRY structures and found highly similar models. We intentionally chose a structure that is the least similar to our model as the starting structure for SAXS-driven MD simulations. Despite the distinct initial positions of the CTE relative to the PHR, the full-length *Cra*CRY converges to structures that possess a molecular shape similar to the ones presented in Fig. 2 (see Supplementary Movie S3). In contrast, starting with our predicted model, conventional MD simulation (without restraints of SAXS data) yielded a structure that is similar to the trRosetta model, whose CTE is closely packed to PHR (see Supplementary Movie S4). This control simulation highlights the importance of SAXS data in the structure refinement of full *Cra*CRY. The proposed structural models are based on information gathered from smFRET experiments, homology modeling, and SAXS-driven MD simulations. These models serve as initial structures that can be validated or improved by further studies, such as high-resolution structure determination experiments.

Blue-light-dependent dimerization behavior of *Cra*CRY was confirmed by single-molecule assay, and dimeric structures of *Cra*CRY consistent with SEC-SAXS data are proposed. The smFRET experiments revealed that *Cra*CRYs with oxidized FADs are monomeric in the dark. Only under blue-light illumination, *Cra*CRY dimerization was observed, and the CTE is closer to PHR in a dimeric state than in their monomeric state, according to the smFRET data (Fig. 4e). Additionally, the SEC assay demonstrated that exposure to blue light can induce the dimerization of the *Cra*CRY PHR domain even without the CTE domain (Supplementary Fig. S3b). This experiment showed that the PHR domain contained the essential interface for *Cra*CRY dimerization, which was consistent with the SAXS data and the constructed structure of dimeric *Cra*CRY (Supplementary Fig. S10b). The conformation and role of CTE have been under intensive investigation. Oldemayer et al. clainmed *Cra*CRY stayed as dimeric states in the 'dark' (absent of red light), and red-light illumination triggers oligomerization. This is in contrast to either Frank-Badur et al. or our work, where *Cra*CRYs exist as monomers in dark conditions. Using Infrared difference spectroscopy based on the absorption of FAD, Spexard et al. did not register any blue-light-dependent conformational change in the CTE domain, which is not contiguous to the cofactor FAD. Frank-Badur et al., on the other hand, proposed that the structured CTE binds to PHR in dark conditions, but it becomes disordered upon light illumination and dissociates from the PHR domain. Here, based on smFRET and solution X-ray scattering experiments, we provide an alternative model, in which the CTE is partially packed to PHR, and the packing pose is influenced by light. It is likely that the FAD oxidation state affects the CTE packing via the regulation of α22 (the helix adjacent to PHR domain), which is in the vicinity of FAD binding site. This light-sensing mechanism is consistent with the model proposed by Frank-Badur et al., although the exact movement of CTE is different. According to SAXS data (Fig. 2b), the dimeric *Cra*CRY or PHR domain are compact, manifested in the peak near the Guinier-Kratky point. There is a slight right shift of the peak position, indicating either flexible conformations of the dimer or a more elongated shape. In the Cryo-EM study of *At*CRY2 proteins, researchers found that the dimeric structure is more dynamic than its tetramer[38]. On the other hand, we noticed that the signal-to-noise ratio (SNR) needs to be enhanced by improving the sample quality and SEC-SAXS experimental protocols. The lower SNR at larger scattering angles for dimers or *Cra*CRY–ROC15 complexes may contribute to the up-shift observed in Kratky plots.

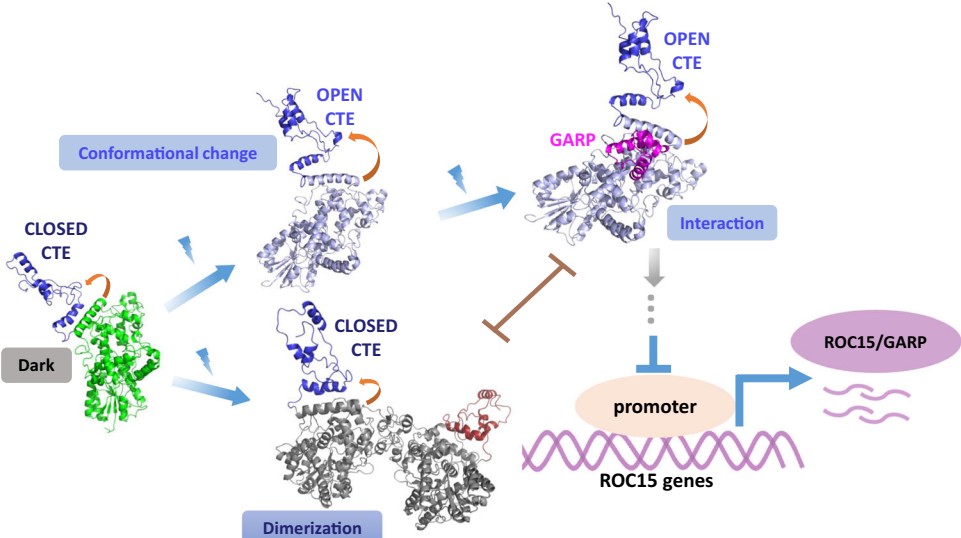

**Fig. 5 Proposed model of the molecular functional mechanisms of *Cra*CRY.** The interplay among blue-light-induced conformational change in *Cra*CRY, dimerization, and interaction with ROC15 are depicted by the arrows. The green image shows the inactivated, closed *Cra*CRY monomer, whereas the blue image shows the monomer *Cra*CRY in the open state. The red and gray images show the monomer units in the dimeric *Cra*CRY. The purple image indicates the GARP motif structure docked to the activated *Cra*CRY. *Cra*CRY structures in monomeric and dimeric states and the *Cra*CRY–ROC15(GARP) complex structure were refined or validated against SAXS profiles.

Another exciting discovery was the light-induced interaction between *Cra*CRY and the GARP domain of ROC15. We identified ROC15 as an interaction partner of *Cra*CRY for circadian rhythm entrainment upon blue-light illumination. Pull-down assays performed on the PHR and CTE domains confirmed that the blue-light-induced interaction between *Cra*CRY and ROC15(GARP) was in the PHR but not the CTE domain (Fig. 3b). The single-molecule pull-down assays further revealed the mutual inhibition between *Cra*CRY dimerization and *Cra*CRY–ROC15 interaction. The single-molecule pull-down assay showed that the fraction of the *Cra*CRY dimers with ROC15(GARP) present in the solution was almost at the background level (from 2 to 7%; Supplementary Tables S4 and S5). Similar results were found for *Cra*CRY–ROC15 complex formation when *Cra*CRY dimerization was allowed (Fig. 4f, Supplementary Table S5). Therefore, we discovered a competitive relationship between *Cra*CRY dimerization and *Cra*CRY–ROC15 interaction. Upon blue-light illumination, the monomeric *Cra*CRYs were activated, enabling their interaction with ROC15 and signal transduction. The rearrangement of the CTE from the PHR domain upon blue-light illumination may facilitate the ROC15 binding. We further compared the ROC15 binding interface with the DNA-binding residues (HHLARH motif [H356–H361] and SQYFR-Y motif [S409–Y415]) for *Cra*CRY's DNA repair function[21]. The docked structures revealed that the DNA-binding site overlaps with the ROC15 binding interface (Supplementary Fig. S10d), suggesting that the DNA repair function of *Cra*CRY may be influenced by dimerization or ROC15 binding.

Previous in vivo transcription-level study of *Cra*CRY has shifted the whole focus of this protein to red-light responses[58]. However, our work on the blue-light-induced functional dynamics of *Cra*CRY has confirmed the conformational change, dimerization, and intermolecular interaction can be initiated with UV to blue light. Especially, ROC15 interacts with *Cra*CRY when the cofactor is in the FADH• state, which is the red-light absorbing state. The neutral radical state has a relatively flat absorption spectrum covering the whole visible light from 400 to 700 nm (Supplementary Fig. S1b). Therefore, excessive blue-light exposure can also promote the transition of FADH• to the fully reduced FADH⁻ state (Supplementary Fig. S11), which is also found in *Cl*CRY4[59]. This may have a profound implication for *C. reinhardtii* where changing levels of blue light or different wavelengths of light in the environment can result in adaptive response.

Based on information obtained from experimental and computational approaches, we propose the following model of the light-induced function of *Cra*CRY at the molecular level (Fig. 5). Under the dark condition, cofactor FAD in *Cra*CRY is in the oxidized state, and the *Cra*CRY is in the compact, monomeric, and inactivated state. Upon blue-light illumination, the cofactor FAD of *Cra*CRY is reduced to FADH•, accompanied by conformational changes and dimerization of *Cra*CRY. Monomeric *Cra*CRY in the lit conformation promotes the interaction with ROC15, triggering downstream signal transduction and entraining the circadian rhythm. When blue-light-activated *Cra*CRY monomer is abundant, they form *Cra*CRY dimers, inhibiting the interaction between *Cra*CRY and ROC15. We speculate that the dimerization of *Cra*CRY can be a protective mechanism, adaptively preventing overresponse to extremely intense light illumination from environments.

In summary, by applying the single-molecule technique, structure prediction, SAXS experiments, and MD simulations, we investigated the blue-light-induced functional mechanism of animal-like CRY in *C. reinhardtii*. We discovered that blue light can induce interactions between *Cra*CRY and the circadian-clock-related transcription factor ROC15, and such interactions were further investigated using in vitro GST pull-down assays and BiFC in vivo. In addition, colocalization single-molecule spectroscopy combined with single-molecule photobleaching counting assays revealed that blue-light-induced intermolecular interaction and partial dimerization of *Cra*CRY are mutually exclusive processes. Notably, this interesting finding was obtained through the precise control and screening of monomeric or dimeric states of the proteins via single-molecule techniques. Based on experimental data, we proposed structure models for *Cra*CRYs in their monomeric and dimeric states and predicted the structure for the *Cra*CRY–ROC15(GARP) complex. These models provide the structural basis for a molecular mechanism of

light-induced signaling through CRY proteins, paving the way for understanding how CRY proteins perceive external activation and relay information to downstream signaling partners.

## Materials and methods

**FRET construct design**. To monitor the conformational changes between the PHR and CTE domains of CRYs, we designed a *Cra*CRY smFRET construct with one of the FRET pairs located in the PHR domain (the native cysteine, C317) and the other at the end of the CTE (the mutant cysteine, E595C). For wild-type *Cra*CRY, there are seven native cysteine residues in the PHR domain—C27, C101, C261, C317, C365, C462, and C482—and no cysteine residue in the CTE. The native cysteine, C365, was kept because it does not have solvent accessible surface area[21], and control experiments confirmed that it would not be labeled by dyes. Therefore, we kept C317 unchanged for site-specific labeling and then mutated the sequence to C27N, C101S, C261A, C462A, and C482A. In addition, we introduced another cysteine mutation (E595C) at the end of the CTE domain for another type of site-specific labeling (Fig. 1b). Before exposure to blue light, it was vital that the cofactor of *Cra*CRY, FAD, was in the oxidized form, which is the form sensitive to blue light (Supplementary Fig. S1). The FRET construct was labeled with ATTO550/ ATTO647 as a FRET pair, and the Förster radius ($R_0$) was 63.0 ± 0.3 Å in our experiment (Supplementary Fig. S12; the reference value given by the ATTO-tec website is 64 Å). More details about the FRET construct design and functional blue-light sensitivity test can be found in Supplementary Table S1 and Fig. 1.

**Protein expression and purification**. Both wild-type (wt) *Cra*CRY and the FRET construct were expressed in *Escherichia coli* BL21(DE3) by using Luria–Bertani (LB) medium (10 g·L$^{-1}$ tryptone, 5 g·L$^{-1}$ yeast extract, and 10 g·L$^{-1}$ NaCl) containing 50 mg·L$^{-1}$ kanamycin for *Cra*CRY(wt) and 100 mg·mL$^{-1}$ ampicillin for the FRET construct. When the optical density at 600 nm (OD600) reached 0.4, the temperature was lowered from 37 to 20 °C for *Cra*CRY(wt), and 18°C for the FRET construct. The induction was initiated using 250 μM isopropyl-b-D-1-thiogalactoside (IPTG) before the OD600 reached 0.6. After 16 h, cells were harvested through centrifugation (5000 rpm, 15 min, 4 °C) and resuspended in 50 mM HEPES pH 7.4 and 150 mM NaCl buffer. The cells were disrupted using an ultrasonic cell crusher (in an ice bath), and debris was removed through centrifugation at 15,000 rpm (4 °C for 1 h). The supernatant was filtered using a 0.22 μm injection filter and applied to a His affinity column. The column was washed with a buffer containing 300 mM imidazole. Every 2.5 mL fraction of the elution from the His affinity column (General Electric Company, GE) was diluted with a five-fold volume of 20 mM HEPES pH 7.4 buffer. The diluted sample was added to a cation exchange column (HiTrap SP HP, 5 mL, GE). After equilibration, the column was washed with 20 mM HEPES pH 7.4 buffer containing 1 M NaCl.

The FRET construct was prepared similarly to the wild-type, but the cation exchange column was switched to an anion exchange column (HiTrap Q HP, 5 mL, GE). All operations during protein expression and purification were performed under dark conditions or under red light if needed (Supplementary Fig. S13). This was to maintain the cofactor FAD in an oxidized state before exposure to blue light. The PHR domain was prepared similarly to *Cra*CRY(wt).

For *Cra*CRY::mCitrine, 500 mL of cells was harvested and resuspended in 50 mM HEPES pH 6.9 buffer containing 5% glycerol and 0.4% TritonX-100. The cells were disrupted using an ultrasonic cell crusher (in an ice bath), and debris was removed using centrifugation at 15,000 rpm (4 °C for 1 h). The supernatant was filtered using a 0.22-μm injection filter and applied to a cation exchange column (HiTrap SP HP, 5 mL, GE). The sample, concentrated to 500 μL, was then purified and transferred to 50 mM HEPES pH 7.4 buffer containing 50 mM NaCl and 5% glycerol. GARP::mCherry cells were harvested and disrupted similarly to those of *Cra*CRY::mCitrine. However, the GARP::mCherry supernatant was filtered using a 0.22-μm injection filter and applied to a His affinity column (GE) first. The column was washed with a buffer containing 300 mM imidazole. Every 2.5 mL fraction of the elution from the His affinity column was diluted with the five-fold volume of 20 mM Tris pH 8.3 buffer. The diluted sample was applied to an anion exchange column (HiTrap Q HP, 5 mL, GE). The sample, concentrated to 500 μL, was then purified and transferred to 50 mM HEPES pH 7.4 buffer containing 50 mM NaCl and 5% glycerol.

For the in vivo BiFC assay, *C. reinhardtii* (*UVM11*) cells were cultured in the dark–light (12–12 h, changed at 9:00 a.m. and 9:00 p.m. Beijing time) photoperiods for 3 days in Tris-acetate-phosphate (TAP) medium at 25 °C, where the concentration of cells was ~10$^6$ mL$^{-1}$. The cells were transfected with *Cra*CRY::mCitrine(N) and ROC15::mCitrine(C) plasmids through electroporation. After antibiotic screening for 2–3 weeks, fluorescence images were obtained using FLIM.

**Blue-light illumination**. A 450-nm LED lamp with a power of 60 mW·cm$^{-2}$ was used for blue-light illumination. The exposure times were all approximately 15 min for smFRET, GST pull-down, single-molecule pull-down, and BiFC.

**Total internal reflection fluorescence microscopy (TirfM) setup**. A high-numerical-aperture oil-immersion microscope objective (Olympus, APON 60×

TIRF, Numerical aperture (NA) = 1.49, Cargille Type DF Immersion Oil) was employed. Proteins were excited by a solid-state 532 nm laser (Coherent, compass 315M-100). A 532 nm dichroic mirror (Semrock FF535-SDi01) was used for reflecting the excitation laser into the objective and separating the fluorescent signal from the reflecting laser. Filter sets for FRET channels were as follows: 649/ LP as a dichroic (Semrock FF649-Di01-25×36) to divide the fluorescence signal into two vertical paths, BP580/60 M (Semrock) in the donor fluorescence signal path, and BP705/100 (Chroma) in the acceptor fluorescence signal path. Filter sets for single-molecule pull-down channels were 550/LP as a dichroic (Semrock FF550-Di01-25×36), BP510-555 for mCitrine (Edmund Optics), and BP705/100 for mCherry (Chroma). An electron-multiplying charge-coupled device (EMCCD, Andor, DU897E) with a dual-view channel splitter (Photometrics, DV2) was used to display two separate channels.

**Fluorescence-lifetime imaging microscopy (FLIM) setup**. A high-numerical-aperture oil-immersion microscope objective (Olympus, PLAPON ×60, NA = 1.42, Cargille Type DF Immersion Oil) was employed. The excitation source was a supercontinuum fiber laser (Fianium ultrafast fiber laser SC-400-4-PP) equipped with an acousto-optical tunable filter (Fianium PX-00027). A 514 nm notch filter (Semrock FF01-512/SP-25) was used for filtering the 514 nm excitation laser, and a 514 nm dichroic mirror (Semrock FF535-SDi01) was employed for reflection of the excitation laser into the objective. The mCitrine fluorescence signal was filtered using bandpass filters (BP510-555, Edmund Optics) and was then collected using a high-speed single-photon counting module (Becker & Hickl HPM-100-40). The resulting photon count was recorded using a counter timer card (Magma ExpressCard/34) controlled by a time-correlated single-photon counting (TCSPC) software suite (Becker & Hickl) to tag photon arrival times using the onboard 40 MHz clock triggered by laser excitation.

**GST pull-down**. His-tag-fused *Cra*CRY(wt)/PHR/CTE and GST-fused ROC15 (GARP) proteins were purified using His and GST affinity columns, respectively. Subsequently, 400 μL of 10 mg/mL *Cra*CRY(wt)/PHR/CTE and 400 μL of 3 mg/mL ROC15(GARP) were incubated together at room temperature in the dark for 20 min. Half of the mixture was injected into a GST affinity column directly in the dark. The sample was diluted using 10 mL of 50 mM HEPES buffer containing 150 mM NaCl and loaded into the GST affinity column. Then, the samples of the mixture before loading (L), the flow-through solution (F), and the elution bound to the GST affinity column (E) were employed for the SDS-PAGE test. The other half of the mixture was illuminated by blue light (450 nm LED lamp) for 5 min and then subjected to the same operations.

**Single-molecule data collection**. All single-molecule fluorescence intensity movies were acquired using TirfM and an EMCCD exposure time of 0.067 s. Upon taking the dead time of the EMCCD (0.033 s) into account, the practical time bin of all the intensity movies was ~0.1 s. We then extracted the fluorescence intensity–time trajectories from the raw movie data for each bright spot with an appropriate signal-to-noise ratio (SNR), 2.5–3.0, as the criterion indicating effective molecules. In the smFRET experiment, the typical intensity–time traces (Fig. 1a) were screened for FRET efficiency analysis. The change points denoted the photobleaching steps of donor and acceptor dyes, and they were identified using the intensity change point method[60]. More details on the FRET efficiency calculation can be found in previous articles[61,62]. In single-molecule pull-down assays, only the number of photobleaching steps was defined and counted for statistics.

**Statistics and reproducibility**. In the smFRET experiment, the numbers of typical intensity–time traces are 13, 22, and 65 for monomer dark, monomer lit, and dimer lit states, respectively. The FRET efficiency value at the one-time interval (set to 0.1 s) contributes a data point to the statistics. Then the mean value and confidence intervals (95%) can be fitted using Gaussian functions. In single-molecule interaction and dimerization assays, the data have been divided into different groups based on the experiment dates, and the statistical standard deviations can be used as error bars. More details have been shown in Supplementary Table S3–S5.

**SEC-SAXS data collection and analysis**. SAXS experiments were performed at beamline BL19U2 of the National Facility for Protein Science Shanghai (NFPS) at the SSRF. The X-ray wavelength λ was 0.9184 Å. Scattered X-ray intensities were collected using a Pilatus 1 M detector (DECTRIS Ltd). The sample-to-detector distance was 2.30 m, thus measuring signals in the resolution range 0.008–0.47 Å$^{-1}$ as defined by momentum transfer ($q = 4\pi \sin\theta/\lambda$, where 2θ is the scattering angle). For high-performance liquid column (HPLC)-SAXS, the buffer used was 50 mM HEPES pH 7.4 buffer containing 150 mM NaCl and 5% glycerol. SEC-HPLC was performed using a Superdex200 column (GE). The Agilent chromatographic system of the BL19U2 beamline of the NFPS at the SSRF[63] was operated at a 0.50 mL·min$^{-1}$ flow rate. The columns and SAXS flow cell were maintained at 25°C. The protein concentration was from 2 to 10 mg·mL$^{-1}$ in the aforementioned buffer. After centrifiltration, 100 μL of sample solution was injected into the Superdex 200 Increase 10/300 GL column (GE). Two-dimensional scattering images were converted into one-dimensional SAXS curves through azimuthal averaging after solid angle correction and then normalized using the intensity of

the transmitted X-ray beam; this was performed using the software package BioXTAS RAW[64]. Background scattering was subtracted using PRIMUS in the ATSAS software package[65]. SASTBX software was used to compute theoretical profiles of atomic structures and fit them to experimental SAXS data[66]. The SAXS data range for model fitting is truncated to proper regions as suggested by the GNOM program of ATSAS software. Low-resolution envelopes of molecules were computed using decodeSAXS[67]. The theoretical profile calculation and envelope model reconstruction were done with default parameters (more details about data collection and analysis are given in Supplementary Table S6).

**Protein structure prediction**. The full-length structure of *Cra*CRY was predicted using a template-based modeling protocol and the global optimization method called conformational space annealing (CSA)[68–70], which was among the best methods in the category of data-assisted protein structure modeling in the Critical Assessment of Protein Structure Prediction 12 (CASP12) experiment[68,71]. The protocol generated protein structure models by optimizing an energy function combined with a chi-square fitting score on the basis of the SAXS profile of *Cra*CRY. The SAXS profile obtained under the dark condition was used for model prediction. Four templates (PDB codes: 3FY4, 2WB2, 4I6J, and 1OWL) were selected by the program to predict the structures. Prior to structure building, the accessory domains or ligands are excluded. Protein complex structure docking was done using ZDOCK 3.0.2 version with default parameters (Supplementary Fig. S14)[52].

**SAXS-driven MD simulation**. We adapted the SAXS-driven MD simulation method developed by Hub and coworkers, using the modified source code of GROMACS 4.6.5[72–74] to bias the conformations toward experimental SAXS data. Prior to SAXS-driven MD simulations, solvated and neutralized systems were subjected to 50 ns of equilibrium MD simulation. Simulation trajectories of TIP3P[75] and ions were also generated. The trajectories obtained from the equilibrium MD simulations were used to define the envelope around *Cra*CRY. Scattering intensity according to the Cromer–Mann parameters[76] of solute atoms was generated using the *g_genscatt* command. MD parameter files were generated in accordance with the instructions at https://biophys.uni-saarland.de/grenoble-tut/index.htm. The trajectories of the solvent and experimental SAXS data were included as input files to initiate the SAXS-driven MD simulations. The scattering signals of momentum transfer q between 0.02 and 0.20 Å$^{-1}$ were used in the SAXS-driven MD simulations. The control simulation following conventional MD simulation protocol was carried out without applying SAXS data restraints for 1000 ns using GROMACS 5.1.2[77]. UCSF Chimera[78] was employed to visualize and analyze the output of the MD simulations.

**Reporting summary**. Further information on research design is available in the Nature Research Reporting Summary linked to this article.

## Data availability
The authors declare that all data supporting the findings of this study are available within the manuscript and its supplementary files or are available from the corresponding authors on request. Source data can be found in Supplementary Data 1.

## Code availability
The softwares for controlling EMCCD (Andor, DU897E), ultrafast fiber Laser (Fianium, SC-400-4-PP), and FLIM microscope (Becker & Hickl, SPCM) are from the original factories. Single-molecular data were analyzed by ImageJ and Matlab. Fluorescence-lifetime analyses were finished by the original software (Becker & Hickl, SPCimage). Lifetime image drawing was using by Matlab. SAXS-driven MD simulation was using the modified source code of GROMACS 4.6.5. VMD and in-house Python scripts were employed to visualize and analyze the output of the MD simulations, respectively. The full-length structure of CraCRY was predicted using a template-based modeling protocol and the global optimization method called conformational space annealing (CSA).

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

## Acknowledgements

P.L. and H.C. contributed equally to this work. We thank Prof. Guoliang Xu and Dr. Jianhuang Xue in Shanghai Institutes for Biological Science (SIBS) for providing algal plasmids and strains. We also thank Dr. Na Li. in SSRF for SEC-SAXS Technical Supports. We would like to thank Jian Chang for his advices on fluorescence imaging and data analysis. We would also like to thank Haw Yang's Lab for their support and discussions. The project is supported by the fundings from National Natural Science Foundation of China (NSFC) (Grant numbers: No. 11274076, 21773039 to Y.-W.T. and No. 31971136, U1930402 to H.L.). Y.-W.T. is also supported by Shanghai Municipal Science and Technology Major Project (No.2018SHZDZX01), SCI & TECH Project (No.20ZR1405800), and ZJLab. This work was also supported by the National Research Foundation of Korea (NRF) grant funded by the Korea government (MEST) (No. 2018R1D1A1B07049312 and 2017R1E1A1A01077717). We also acknowledge Center for Advanced Computation (CAC) of Korea Institute for Advanced Study (KIAS) for providing computing resources (Linux Cluster System). This research work is supported by a Tianhe-2JK computing time award at the Beijing Computational Science Research Center (CSRC).

## Author contributions

P.L., H.C., Y.-W.T., and C.M. designed the experiments. P.L. and H.C. performed all the single-molecule fluorescence experiments, SAXS data acquisition, GST pull-down assays, and protein purifications. P.L., H.C., and C.M. carried out the CraCRY plasmid mutation. H.C. and C.M. contributed western blotting experiments. Protein structure prediction were performed by K.J. and J.L. The SEC-SAXS data analysis and SAXS-driven MD simulation were carried out by H.L., V.K., C.S.L., X.L., and Y.S. Finally, P.L., Y.-W.T., H.C., H.L., and J.L. wrote the manuscript with inputs from all other authors.

## Competing interests

The author declares no competing interests.
