## [Peer Review File · Communications Biology]

Reviewers' comments:

Reviewer #1 (Remarks to the Author):

Please see attached document for comments.

Reviewer #2 (Remarks to the Author):

The manuscript "Direct Experimental Observation of Blue-Light-Induced Conformational Change and Intermolecular Interactions of Cryptochrome" is a powerful solution-state study leveraging the strengths of FRET, SAXS and advanced MD modelling to understand the structure and mechanism of change undergone by light sensitive receptor proteins. The manuscript provides structural insights on CraCRY, the *C. reinhardtii* ortholog involved in blue-light mediated gene transcription and DNA repair. The authors present in vitro pull-down and in vivo bimolecular fluorescent complementation assays to identify direct binding partners of CraCRY and also characterize dimerisation state of CraCRY using solution-state modelling and analytical size-exclusion chromatography. The authors show a strong role of the PHR domain that it is involved in mediating interactions with ROC15 and responsible for the observed CraCRY dimerization. I appreciate the correspondence between the FRET studies revealing ~15 Angstrom change that is supported by the SAXS-MD modeling. I recommend the manuscript be revised.

The experiments are thorough, well supported however, some issues need to be address prior to publication. The use of Chen and Hub's SAXS-directed molecular dynamics approach to refinement is outstanding. This is exactly how structures should be refined when using SAXS data as opposed to the blind conformational space searching that is typically employed when using CRY SOL or FOXS. As Chen and Hub demonstrated in their seminal paper, MD is incapable of transitioning to the correct states without the added experimental data incorporated during the gradient descent. However, these algorithms will only work well if, 1) the SAXS data represents a single conformational state and 2) the SAXS data is free from instrumentation or measurement artefacts. The authors have addressed 1) by performing inline SEC-SAXS to provide optimal background subtraction as well as resolution of oligomeric species. Point 2 is harder to discern but a major contributor to it would be capillary fouling or aggregation during X-ray exposure.

In the model-independent characterisation of the conformational changes using FRET, the authors were able to show a change from 50 to 65 Angstroms between residue C317 and E595C when exposed to blue light. This can be interpreted as a compaction of the protein with respect to the analytical SEC (Supporting Fig 3a) where dark-state CraCRY elutes at a slight longer time than lit-state. This model independent analysis could be extended further to the SAXS data as the radius-of-gyration should show a notable change between the lit and dark states of monomeric CraCRY as well as d_{max} , and the corresponding $P(r)$ -distributions. An important assessment to make when performing molecular modelling is if the data could/should be represented by a single conformation. The dimensionless Kratky plot is an excellent tool to show how the thermodynamic state of the protein changes between the dark and lit states and I would recommend that a plot using data from Figure 2 be made with the dimensionless Kratky plot and $P(r)$ -distributions normalised to particle volume. Dimensionless Kratky requires R_g and I_{Zero} to be determined and if the lit-state shows a peak shift towards the upper right quadrant, then it likely suggests the protein is becoming more flexible and that the fitting may require more than one conformation. Its hard to discern the quality of the fits in Fig 2, but by visual inspection, the lit-state is not described in low- q by the model, the R_g of the data is much larger than the model. Residual plots for all fits should be shown, the chi-statistic is hiding the poor fit in the low to medium q region.

The modelling of the dimeric form of CraCRY is not supported by the SAXS data. Again, a residual plot will show this with Figure 4a, the low-chi is an artefact of the fit as it is clear from the figure, the fit

oscillates and the chi-statistic is largely driven by the magnitude of the errors in the fit. The PHR dimer model-data fit is more believable.

Again, with respect to the CraCRY-GARP complex (SFig5), the fit is not accurately described, the chi-statistic is being driven by the noise and in this case, the authors fit the entire q-range - partly to drive down the chi-statistic which is misleading. I would suggest a dimensionless Kratky plot be made of the lit and dark states of monomeric CraCRY, GST-GARP+CraCRY data in SFig5, and of the data in SFig4. It may be revealing to show that the binding of GST-GARP resembles the monomeric lit state suggesting binding does not change the compacted state. It is clear from the fit, the low-q region is not explained by any part of the model.

The data was collected to 0.47 inverse Angstroms, yet was truncated to 0.25 in the presentation in Figure 2, extended slight more in SFig4b and then used entirely in SFig5? Some explanation is required in the text. In some cases, it may be that the data isn't fully transformable to real-space due to poor background matching or capillary following and is truncated to support a P(r)-distribution. The danger of using the noise to higher q-values is it artificially reduces the chi-statistic and this is why residuals are a better guide to confirming the quality of the fit. The residuals should be randomly distributed without oscillating features.

Minor Comments/Questions:

Figure 1 Legend "The molecular weight is 70195 daltons, very close to the theoretical value at 70195.7 daltons" could be shortened to "ESI determined molecular weight is 70195 (theoretical 70195.7) daltons."

Figure 3A, UV trace for CraCRY Dark appears to go under the baseline or looks truncated whereas in 3b, the traces clearly above the baseline. I would appreciate it if the authors can show the baseline for the UV trace more clearly in the figure.

In Supporting Figure 4b, how many points are negative? A $q \cdot I(q)$ vs q of the fit would be good to show, however, it would appear that the data above 0.2 inverse angstroms carries little information.

From Materials and Methods, "Low-resolution envelopes of molecules were computed using decodeSAXS". I did not see any presentation of low-resolution envelopes. Not relevant to the study.

Was the FAD molecule included in the MD simulations? It is not apparent in the supporting movie.

Revision to the reviewers -- COMMSBIO-21-0228-T

We appreciate the time and feedbacks from both reviewers. We carefully and thoroughly revised the manuscript to incorporate the suggestions and to address the concerns. Several control experiments and analyses were conducted to validate or better support the conclusions. The results presented in the original manuscript were further substantiated with these additional data.

In the revised manuscript, major changes are highlighted in **blue color**, and we prepared a separate Table-of-Change to list major updates.

Revised figures are included in this document for quick reference (these figures are listed with a prefix R, such as Figure R1, R2...), and their corresponding figure numbers in main text or supporting information are included in figure legends.

Reviewer #1:

In Li et al, the authors examine light-induced oligomerization, and conformational changes in the animal-like CRY from *Chlamydomonas*. They employ a combination of SAXS and MD-based modeling to construct putative structures of light- and dark-states of *CraCRY*. Further, they report that they identified a protein interaction partner ROC15, that they validate using SAXS, and a variety of *in vitro* and *in vivo* approaches. **Overall, the data is intriguing and is an interesting approach**, however the overall data and methods are significantly lacking in details that complicates analysis of the manuscript. Based on the data presented, I have significant concerns on whether their conclusions are fully supported by the data. Additional data and expanded methods are necessary for a proper evaluation. Below are both major and minor concerns that need to be addressed.

Reply:

We thank the reviewer for detailed comments and the complement on the intriguing data and the interesting approach. We revised the manuscript accordingly and addressed the concerns, especially concerns on whether the conclusion is supported by the data. In the article, the single-molecule assay data alone (single-molecule FRET/single-molecule photobleaching screening assays) provide strong supports to the following main conclusions:

- 1) blue light induces conformational change and homo-dimerization of *CraCRY*;
- 2) blue light induces *CraCRY*-ROC15 interaction, which were also confirmed by GST pull-down *in vitro* and BiFC *in vivo*;
- 3) the mutually exclusive relationships between “*CraCRY*-ROC15 interaction & *CraCRY* homo-dimerization”; and
- 4) the CTE is closer to PHR in *CraCRY* dimers.

These observations are supplemented by structural analysis using computational prediction, SEC-SAXS, and molecular dynamics simulations.

As mentioned in the manuscript, the CTE domain of *CraCRY* is presumably highly flexible and therefore difficult to crystallize and resolve its structure to atomic resolution (Franz, S. et al, 2018, *Nucleic Acids Research*). That motivated us to apply **SAXS experiments and MD-based modeling** to acquire **plausible 3D models of the full-length *CraCRY* in dark and lit conditions**

and subsequently **the *CraCRY*-ROC15 complex models**. These 3D models provide valuable insights and support the main discoveries from single-molecule assays.

In the revised manuscript, we included additional data and analysis results suggested by the referees, especially the detailed analysis on SAXS data and model building that were not presented in the original manuscript. Nicely aligned with the results in the original manuscript, these new analyses enhanced the supports to our conclusions.

Below, we include the detailed point-to-point replies:

Major Issues

A) One of the main claims of the manuscript is that they “obtained dark- and lit structures of full-length *CraCRY*, including the CTE.” The claim is difficult to support with the given methodology and data presented.

Their structures are based on SAXS and molecular modeling approaches, and has inherent flaws, further complicated by the lack of a full presentation of SAXS data needed to evaluate the validity of the results. I address these below:

1) The authors indicate that they used SEC-SAXS to be able to isolate the monomeric fraction of the sample. They reference Supplemental Figure 3 as the SEC traces, however those appear to be conducted using different flow rates than the SAXS experiments, so they are likely not the actual SEC-SAXS traces.

Further, they do not provide the SEC trace, and corresponding sample region they used for SAXS analysis. That should be provided, and they should also provide the corresponding predicted MW traces for the peak/volume selection. This is imperative in determining how mono-disperse the actual sample was. Those curves are readily available from the primary data analysis in BioXTAS RAW.

Reply:

The SEC plots shown in Supplemental Figure S3 (Fig. S3) were indeed adopted from the stand-alone SEC experiment for clarity. The curves shown in the original Fig. S3 were the SEC traces obtained on an FPLC not in line with the SAXS beam, as explained in the caption, the flow rate for that experiment was 0.8-1.0 mL·min⁻¹ (updated in page 36). The flow rate used in SEC-SAXS experiments was 0.5 mL·min⁻¹ (page 17), different from the stand-alone SEC experiments. The standard SEC data in the original Fig. S3 was used to emphasize the blue-light induced dimerization of *CraCRY*, not to illustrate the SEC-SAXS experimental data.

Thanks for the suggestion and to eliminate potential misunderstandings, in the revised Supplemental Figure S3, we added the corresponding SEC-SAXS traces (Fig. S3c, d). The regions centered around peak positions were used to obtain scattering intensities of dimeric-/ monomeric *CraCRY* or PHR domain (indicated by the shaded areas).

Using the BioXTAS RAW program, we calculated the radius of gyration (R_g) and molecular weights (M_w) from SEC-SAXS profiles. The UV absorption signals, integrated SAXS intensities, R_g/M_w for full-length *CraCRY* under lit condition is shown in a new supplementary figure (Fig. S4, also shown as Fig. R2 for quick reference), illustrating correspondence between UV absorption signals and SAXS data.

Figure R1 (updated Supplemental Figure S3). SEC traces obtained by stand-alone SEC experiments and SEC-SAXS experiments. a. SEC curves of *CraCRY* under dark and lit conditions obtained by SEC experiments. The dotted line is the calibration curve from the mixture of monomeric and dimeric BSA proteins, of which the molecular weight is about 66 kDa, very close to *CraCRY*(wt) (70 kDa). Only one peak appears at 13.6 mL under dark condition (black line), while two peaks at 11.8 mL and 13.2 mL appear under lit condition (blue line). The new peak at 11.8 mL corresponds to the dimer of *CraCRY*. **b.** SEC curves of PHR domain under dark and lit conditions obtained by SEC experiments alone. Similar to *CraCRY*, compared with dark curve (black line), the extra peak in lit curve (blue line) shows the PHR domain can also be dimerized upon blue-light illumination. SEC was performed at room temperature with Superdex 200 (GE). The flow rate is 0.8–1.0 ml·min⁻¹. **c.** SEC and scattering intensity curves of *CraCRY* under dark and lit conditions obtained by SEC-SAXS experiments. **d.** SEC and scattering intensity curves of PHR domain under dark and lit conditions obtained by SEC-SAXS experiments. In SEC-SAXS experiments, the elution times were recorded instead of elution volumes and the absorption signals (at 280 nm, black lines under dark conditions and blue lines under lit conditions) were saturated since the protein concentrations were very high. The scattering intensities of *CraCRY* and PHR domain under dark (grey line) and lit (orange line) conditions are also shown respectively. And the regions marked by grey and orange blocks correspond to the samples used for SAXS analysis. The grey regions were for SAXS analysis on monomers, while the orange regions correspond to dimers.

Fig. R2 (Fig. S4 in the revised manuscript). The series analysis plot for *CraCRY* SEC-SAXS experiment. The X-axes correspond to the flow-through time in SEC assay (top) and the frame number during SEC-SAXS experiment (bottom). The black and blue solid lines represent the absorption of sample at 280 nm and the integrated X-ray scattering intensities, respectively. The platform of absorption trace near 26-27 min appeared because the threshold of detector was exceeded. The gray area near 250 frame is the signal of dimeric *CraCRY* and near 325 frame is that of monomeric *CraCRY*. Green and orange dots exhibited the estimated Radius of gyration (R_g) and molecular weight (M_w) values for both monomeric and dimeric *CraCRY* proteins.

- 2) It is expected that SAXS data will include Guinier analyses, pairwise distribution profiles, and the corresponding D_{max} used to construct any downstream results. None are provided here making it impossible to assess the overall data quality.

Reply:

We thank the reviewer for this constructive suggestion. The basic analysis results of SAXS data are now included in the supplementary materials (the supplementary Note, due to the large size of Tables). Following the guidelines by the SAXS community (Trehwella et al. Acta D, 2017), we compiled a table to summarize the SAXS experimental information and parameters from basic analyses, such as Guinier analysis, $p(r)$ fitting, etc. The data-model fitting figures are also updated by including residual plots for all fitting results (Fig. 2 and Fig. S6).

- 3) Direct overlays of the light vs. dark pairwise distribution profiles, and Kratky plots are also typical and expected to assess the magnitude of conformational changes. The changes in structure in Fig. 2C are not large (largely differ in D_{max}), and I have serious concerns on whether the dark- and light- models can be differentiated based on the data quality shown in Fig. 2a and 2c. It would help to possibly include the full data range for q . The data presented right now is truncated to 0.25 but was collected to 0.47 \AA^{-1} .

Reply:

The D_{max} of lit state *CraCRY* is about 7 \AA larger than the value of dark state protein, according to the pairwise distance distributions, and the $p(r)$ profiles are included in the revision (Fig. 2g). The PHR domain of *CraCRY* is stable throughout the molecular dynamics refinement ($<2.3 \text{ \AA}$ RMSD with respect to the initial structure), while the CTE displays substantial conformational changes. Since the PHR domain is the larger domain with about 83% of amino acids (495 out of 595 residues), the differences in the SAXS profiles (as well as the smFRET signals) are originated from the conformational changes of CTE and its relative movement with respect to the PHR domain. According to the SAXS-driven MD simulation results, we found that

CTE domain exhibits ~ 22 Å movement in the lateral direction, after aligning the PHR domains.

In this revision, we showed the SAXS profiles with measured data range up to $0.47/\text{Å}$ in Fig. 2a. The truncation to q range is by considering the SAXS data quality, specifically, the signal-to-noise ratio is too low beyond $0.25/\text{Å}$ in most cases. In the revised manuscript, the data fitting is carried out for the recommended q -ranges by Gnom $p(r)$ analysis, and we added truncation criteria explanation to the Method section (updated in Page 17 of the revised manuscript):

“The SAXS data range for model fitting is truncated to proper regions as suggested by the GNOM program of ATSAS software.”

- 4) The authors indicate that low-resolution envelopes were computed. That data should be provided and compared to the predicted light vs. dark structures.

Reply:

The low-resolution maps were computed and aligned to the refined atomic models, but omitted in the previous version. These results are now included in the revised manuscript (Fig. 2d). The reconstructed low-resolution envelope models also suggest a movement of CTE induced by blue light illumination. Thanks to the reviewer's suggestion, the low-resolution envelope models in the revision enhance the argument of light induced arrangement of CTE.

Fig. R3 (updated Figure 2). SAXS data and modeling. a. SAXS data in the range of measurement. The intensities were scaled for presentation purposes. b. Dimensionless Kratky plots for the SAXS data. c. Structure ensembles obtained from SAXS-driven MD simulations. d. Reconstructed low-resolution models superposed to the crystal structure of PHR and the representative models of CraCRY in dark and lit states, shown in panels from left to right respectively. e. The theoretical SAXS profiles (solid red) calculated from the atomic models, fitted to experimental data (black dots). The residual plots $(I_{\text{expt}} - I_{\text{model}}) / \sigma_{\text{expt}}$ are shown underneath each fitting result. f. The positions of CTE in representative structures, with dark state structure shown in grey color, and the lit state in blue (see d). g. The pairwise distance distribution function $p(r)$ derived from SAXS data in dark and lit states.

- 5) The structures presented are computed using approaches that do not work well for unstructured regions, where they will bias them towards a structured conformation. The CTE of CRYs are typically unstructured, thus the resulting predicted structure of the CTE would be expected to be a very poor approximation of the actual structure. Indeed, their CTE structure is largely folded, and displaced from the main PHR. Both are likely

artifacts of the computational approach.

Reply:

The 3D structure prediction algorithms may bias towards structured conformations, but the secondary structure predictions are often reliable. Therefore, we carefully examined the predicted structures of CTE domain, and found that the four helical segments are consistent with the secondary structure prediction results, especially the helix connected to the PHR domain and the helix near the C-terminal (Fig. S5 in the revised manuscript). The sequences at the other two helical regions also show strong helical propensity. For the structures presented in this work, the CTE is mostly composed of unstructured regions, the predicted and refined structures have about 37% (+/-5%) helical conformation as structured regions. The majority of the CTE does not possess ordered structures, which is consistent with expectation. Furthermore, the predicted structure shows a CTE packed onto the PHR (like most globular proteins, Fig. 1b), but the structure becomes less compact as the SAXS-MD refinement progresses, with the CTE partially dissociated from PHR domain (we illustrated this in the supplementary simulation movies). Specifically, the CTE moves away from PHR, accompanied by the local unfolding of the predicted structure. Structure prediction may bias towards structured conformations due to the algorithm design, but the conformational changes observed in SAXS-driven MD simulation show that this approach is useful in rectifying such bias.

Importantly, the structure they present is inconsistent with all other data on the *CraCRY* CTE. First, the Kottke lab saw no changes in secondary structure in the CTE upon photoexcitation using FTIR approaches (<https://doi.org/10.1021/bi401599z> and <https://doi.org/10.1074/jbc.M116.726976>). The models presented here have clear changes in helical elements that would be readily determined by FTIR. Second, using HDX-MS, Franz-Badur et al. demonstrates that the CTE is likely unstructured and packs against the PHR domain. These results further suggest that the dark- and light-state structures presented suffer from artifacts from the computational approach.

Significantly more detailed SAXS data and analysis is needed to support their assertions on the structures.

Reply:

The predicted and the refined structures show that the CTE has about 37% helical structure on average and the rest do not have ordered structures (i.e., loops and turns). From the structure ensembles obtained via MD simulations, we found 38.6% and 36.7% of helical conformations in the CTE at dark and lit states, respectively. In other words, more than 60% of CTE residues are unstructured, which is consistent with the HDX-MS results mentioned by the reviewer. Although we cannot completely rule out model bias of the prediction algorithm, the SAXS-driven MD simulations help reduce the potential bias (supported by supplementary movies). Refined structures are well fitted to the low-resolution envelope models that are reconstructed from SAXS data (updated Fig. 2). Given the SAXS data, smFRET signals, and the structure prediction/refinements, we present our best efforts in determining plausible models to better understand the blue-light induced molecular interactions and the underlying molecular mechanism of signal transductions. These models can be validated or improved when high resolution structural information becomes available. We included a statement on the structures in the revised

discussion (page 11):

“The proposed structural models are based on information gathered from smFRET experiments, homology modeling, and SAXS-driven MD simulations. These models serve as initial structures that can be validated or improved by further studies, such as high-resolution structure determination experiments.”

Kottke lab first applied a blue-light pre-illumination, then the experiment is carried out under red light excitation. On the contrary, in our experiment, **we focused on the conformational changes of *CraCRY* upon blue-light illumination, without the following red-light illumination.** Kottke lab did not observe secondary structural changes in the CTE upon red light excitation using FTIR approaches. This is not contradictory to our models, since the major structural changes is due to the rearrangement of CTE, whose secondary structures are not changed significantly. In 2019, by applying high temporal resolution of TRXSS method, Oskar Berntsson et al. found a coordinated change of interatomic distances around 6 Å of full-length *DmCRY* upon blue-light illumination and the change is absent in XIPho that does not contain the CTT. ‘Considering the pitch of α helices of 5.4 Å, the signal would be consistent with structural changes of the partly α -helical CTT (*Sci Adv.* 5 (2019), doi:10.1126/sciadv.aaw1531).’ This implies the light-induced structure changes of CTE in *DmCRY*, similar to our results in *CraCRY* with/without blue-light illumination.

Moreover, Franz-Badur et al. demonstrates that the $\alpha 22$ helix (H475-K494, the last helix of PHR domain linked to CTE) of *CraCRY* likely packs against the body of PHR domain (D32-D323) in dark. And upon blue-light illumination, the H-bond network between $\alpha 22$, D323 and D321 will be disrupted, resulting a movement of the $\alpha 22$ helix away from PHR domain and therefore a change in the CTE position. These are consistent with the SAXS-driven MD simulation results, in which the dark state CTE is partially packed to PHR domain, and the packing becomes looser under blue-light illumination.

For clarity, we included detailed SAXS data analysis results in the revised manuscript. We also carried out more extensive SAXS-MD simulations to refine the structural models under the constraints of SAXS data. As a comparison, we applied Rosetta method to predict alternative models. The secondary structures of the CTE are highly consistent with our predictions. Among the five best models, one model is very similar to the model presented in the manuscript. We picked another model whose CTE is packed to a different region of PHR, and used it as the initial structure for SAXS-driven MD refinement. Strikingly, the refinement showed that the CTE domain underwent even more substantial rearrangement and finally converged to the position that resembled our refined models presented in the article. We compiled the simulation into a video to show the process (included in the supplementary materials, Movie S3). The converged results strongly enhanced the confidence of our structural models of *CraCRY*.

B) A second major claim of the paper is the determination of a new protein-protein interaction target for *CraCRY*, and corresponding examination of the role of CRY homo- and CRY-ROC15 hetero-dimerization. Although, in general their data is compelling there are two potential major issues.

1) All the data collected in examining both homodimerization, and the CRY-ROC15

heterodimer was done in non-reducing buffers. Both Oldemyer et al., and Franz-Badur et al., clearly demonstrated that homo-dimerization is dictated by disulfide bond formation. This is problematic because it indicates that *CraCRY* has surface reactive Cysteines, and the CRY photocycle is known to generate ROS that catalyze non-specific disulfide bond formation. This leads to non-biological oligomerization in drosophila CRY (<https://doi.org/10.1016/j.chemphys.2007.09.014>), and leads to biologically relevant disulfide bond formation in Arabidopsis CRY2.

The CRY homodimerization analysis must be done in the presence and absence of a reducing agent (TCEP/DTT).

Given that ROC15 is almost entirely comprised of low-complexity regions (over 90% predicted to be unstructured), with multiple Cys residues, it is also highly likely to form non-specific disulfide mediated dimers in the presence of ROS. That could lead to light-specific artifacts in CRY interactions. The experiments need to be done in the presence of reducing agents to validate that the interaction is specific.

Reply:

To our knowledge, the aggregation of *DmCRY* discussed in <https://doi.org/10.1016/j.chemphys.2007.09.014> was not attributed to non-specific disulfide bond formation. It is proven that *DmCRY* is monomer in solution in either dark or lit conditions by *nonreducing* SDS–polyacrylamide gel electrophoresis (DOI: 10.1126/sciadv.aaw1531). On the other hand, no disulfide bond was found in crystal structure of *AtCRY2* tetramer (PDB: 6X24).

For *DmCRY*, there is no discussion that non-specific disulfide bond formation can lead to non-biological oligomerization in the reference paper (<https://doi.org/10.1016/j.chemphys.2007.09.014>). On the contrary, *DmCRY* in solution is monomer in dark and lit proved by nonreducing SDS–polyacrylamide gel electrophoresis (DOI: 10.1126/sciadv.aaw1531). W374A mutant of *AtCRY2* showed similar oligomerization with lit state by SEC experiments, which means oligomerization was not induced by ROS (<https://doi.org/10.1038/s41594-020-0449-x> and <https://doi.org/10.1038/s41477-020-00800-1>). And no disulfide was found in crystal structure of *AtCRY2* tetramer (PDB: 6X24).

According to our experimental results, for *CraCRY*, disulfide bonds formation is not the cause of either blue-light induced dimerization or its interaction with ROC15. *CraCRY* dimerization and *CraCRY*-ROC15 interaction are closely related to the redox states of FAD, which is influenced by the blue light. The contrast experiments under dark or lit conditions clearly demonstrated the underlying relations. As suggested, control experiments were carried out to validate this conclusion.

Both Oldemyer et al., and Franz-Badur et al. found the blue-light induced dimerization of *CraCRY* would be suppressed with TCEP or DTT, the reducing reagents that protect Cysteines from forming disulfide bonds. At a first look, it appears that the dimerization was due to the disulfide formation, if experiments were conducted without reducing reagents. However, we noticed that the FAD redox state is also influenced by the reducing reagents, and the redox state of FAD is the real cause of the suppressed *CraCRY* dimerization. We carried out *CraCRY* homodimerization analysis in the presence of TCEP or DTT (Figure R4, a-d). The results indicate that there is indeed no dimerization peak appears with TCEP/DTT (Figure R4, a-c), consistent with Oldemyer et al. and Fanz-Badur et al.. We also measured the absorption spectra, which verified that the FAD cofactors were further reduced to FAD⁻ state even under blue-light

illumination (Figure R4d), which is different from the usual FADH state in the absence of DTT (Fig. S1b). The difference in FAD redox states result in different surface charge distributions, which may affect the dimerization of *CraCRY*.

With the above observation, we believe that the reducing reagents not only protect cysteine from oxidization, but also influence the redox state of FAD. To test this hypothesis, we carried out a control experiment using Iodoacetamide (IAM) to block the Cysteines (Tremblay JM at el. *Biochemistry*. 40, 9151–9158 (2001)) on the surface of *CraCRY* (**Figure R5a**), and then repeated the homodimerization analysis of *CraCRY*. This time, the results indicate that the blue-light induced dimerization was not affected by Cysteines-blocked *CraCRY* (**Figure R5c**), and the photoreduction cycles of Cysteines-blocked *CraCRY* (**Figure R5b**) are similar to the case without IAM (**Fig. S1b**). These results clearly prove that the blue-light induced dimerization is not related to the disulfide bonds formation.

Regarding ROC15, although ROC15 has multiple Cysteine residues, the GARP motif in ROC15 used in our GST pull-down assays does not have any Cysteine residues (**Figure 3a, b; Figure R6a, b**). The positive results in GST pull-down assays of ‘*CraCRY*(wt) vs ROC15(GARP)’ and ‘FRET construct vs ROC15(GARP)’ proved that the *CraCRY*-ROC15(GARP) interaction is independent of disulfide bonds formation. Combined with the negative results between GST tag and *CraCRY*(wt) (**Figure R6c; Fig. S7b**), we conclude that the interaction between ROC15/ROC15(GARP) and *CraCRY* was not due to non-specific disulfide bond formation.

Similar to the case of dimerization experiments, we repeated the GST pull-down assays between ROC15(GARP) and *CraCRY* with 50 mM DTT or 10 mM TCEP (Figure R4e, f). No obvious interactions have been observed when the FAD was further reduced to FAD⁻ state by DTT/TECP upon blue-light illumination. However, the interactions between ‘Cysteines-blocked-by-IAM PHR domain’ and ‘ROC15(GARP)’ were reproduced, similar to the dimerization experiment (Figure R5d). All these evidences support that the redox states of FAD dictate the blue-light induced dimerization and interactions of *CraCRY*, instead of the non-specific disulfide bonds between protein molecules.

Fig. R4. Blue light dependent dimerization of *CraCRY* and PHR suppressed with TCEP and DTT.

a. SEC (green solid) and scattering intensity (orange-yellow solid) curves of *CraCRY* upon blue-light illumination with 10 mM TCEP. While, the dotted lines are SEC (skyblue) and scattering intensity (brown) curves of *CraCRY* upon blue-light illumination without TCEP. **b.** SEC (green solid) and scattering intensity (orange-yellow solid) curves of PHR domain under dark and lit conditions with 10 mM TCEP. While, the dotted lines are SEC (skyblue) and scattering intensity (brown) curves of PHR domain blue-light illumination without TCEP. **c.** SEC (green solid) curve of *CraCRY* blue-light illumination with 50 mM DTT. While, the skylight dotted line is without DTT. **d.** Photoreduction of purified *CraCRY* with 10 mM DTT. These absorption spectrums were monitored by UV-VIS spectrometer. Before blue-light illumination, the *CraCRY* proteins stay at FAD oxidized state (black line, Dark with DTT). With 10s blue-light illumination, the protein will be partially reduced to FADH state (red line, BL 10s with DTT). And then, *CraCRY* is further reduced to FAD- state after 120 s blue-light illumination (cyan line, BL 120s with DTT). This process is partly reversible. Part of the reduced protein will return to oxidized state after remaining under dark conditions for 20 min (brown).

e-f. Pull-down molecular complex analyzed using SDS-PAGE. Green arrows mark the *CraCRY*(wt). Pink arrows mark the band of GST-ROC15(GARP). The images show that *CraCRY*(wt) can not be eluted from the GST column under both dark and lit conditions in the presence of DTT/TCEP. It is worth noting, since the blue-light response in absorption spectrum of *CraCRY*(wt) also changed within DTT/TCEP. The negative interaction results here due to the different spectral response of FAD cannot be ruled out.

Fig. R5. Blue light induced dimerization of Cysteines-blocked *CraCRY* and interaction with ROC15(GARP) of Cysteines-blocked PHR domain. **a.** Cysteines on the surface of *CraCRY* blocked by IAM. Ellman's Method (DTNB) has been used to measure surface reactive Cysteines of *CraCRY* and the ratio of absorbance at 412 nm to 280 nm indicates the average number of cysteines in every protein. **b.** Photoreduction of purified Cysteines-blocked *CraCRY*. **c.** SEC curves of Cysteines-blocked *CraCRY* under dark and lit conditions. **d.** PHR vs ROC15(GARP) interaction complex pull-down displayed by SDS-PAGE. Green arrows mark the band of Cysteines-blocked PHR domain. Pink arrows mark the band of GST-ROC15(GARP). The left image shows that both ROC15(GARP) and PHR can also be eluted from the GST column in blue light dependent manner, showing that the Cysteines-blocked operation does not eliminate the interaction between ROC15(GARP) and PHR domain. In other words, the interaction between ROC15(GARP) and PHR domain should not be dependent on the formation of disulfide bonds.

Fig. R6. Amino acid sequence of ROC15(GARP) and the GST control experiments by pull-down assay. **a.** Sketch map of protein designs for ROC15(GARP) in GST pull-down assay. **b.** Amino acid sequence of ROC15 (black and pink) and ROC15(GARP) (pink part). **c.** GST vs *CraCRY*(wt)

interaction complex pull-down displayed by SDS-PAGE. Green arrows mark the band of *CraCRY*(wt). Pink arrows mark the band of GST. The image shows there is no interaction between GST and *CraCRY*(wt), demonstrating that the *CraCRY*/PHR/FRET construct can only be baited by ROC15(GARP), not the GST tag.

- 2) The in vivo data is compelling, however as the authors note, Chlamy will generate significant auto-fluorescence, and the proteins being overexpressed impact circadian rhythms. An examination of their data shows significant variation between overexpresser lines and the WT, and significant variations between light and dark even for WT. This suggests some light or protein specific variation in the autofluorescence. The only data with both over-expressed is the BiFC data, but no *CraCRY*:ROC15 double overexpressing control was done (e.g. no mCitrine), this should be done to make sure there isn't some other pigment impacted by overexpression of both species that leads to a different auto-fluorescence signature.

Reply:

The *CraCRY*:ROC15 double overexpressing control is supplemented in the revision. The results showed that both the intensity images and lifetime distributions under dark and lit conditions (Figure R7, included in the supplementary Figure S8) were consistent with singly transfected *C. reinhardtii* cells (Figure 3g). The data showed that there is no other pigment affected by overexpression of both species that leads to a different auto-fluorescence profile.

Fig. R7 (Fig. S8 b, c) . Fluorescent images and lifetime distributions of *CraCRY:ROC15* double overexpressing control. a. Fluorescent images under the dark (left) or lit (right) condition. **b.** Grey and blue lines indicate the lifetime distribution for each image under the dark and lit conditions, respectively. The green arrow points to the characteristic lifetime component of mCitrine signals, used to indicate whether an interaction has occurred (**Figure 3g**). Since the difference is mainly between 800 and 3200 ns, only this part of data is shown here to make the comparison more obvious and consistent with the main text.

They also say the data was monitored throughout the circadian cycle, but do not provide any data on the circadian variance. This should be provided.

Reply:

Circadian cycles in protein expression levels of *CraCRY* and *ROC15* have already monitored by Beel B. et al. and Niya Y., et al. by SDS-PAGE and bioluminescence assays respectively (Fig. R8a, b). For completeness, we added our data of doubly transfected *CraCRY*-mCitrine and *ROC15*-mCherry expression cycles (Fig. R8c, included in the Supplementary Fig. S8). In terms of the intensity changes, all of the data show that the expression levels of both *CraCRY* and *ROC15* follows the rule of accumulation in dark condition and decrease in light condition.

Fig. R8. Circadian cycles in protein expression levels of CraCRY and ROC15. **a.** Expression cycles of CraCRY. *C. reinhardtii* wild-type cells were grown under a LD12:12 cycle and harvested at the indicated time points (Beel, B et al, *Plant cell*, 2012, 24, 2992-3008). **b.** Expression cycles of ROC15 by Bioluminescence assays (Niya, Y, et al, *PNAS*, 2013, 110, 13666-13671). **c.** (included in the supplementary Fig. S8). Our results of expression cycles of doubly transfected CraCRY-mCitrine and ROC15-mCherry monitored by fluorescence imaging. *C. reinhardtii* cells were grown under a LD12:12 cycle (L and D are short for Lit and Dark respectively, and LD means the Lit and Dark alternating cycles) and imaged at the indicated time points. The bar below filled with black from LD14 to LD22 means cells incubated in dark condition, while unfilled part means in lit condition. The number following LD means the time point during the 24-hour cycle (unit: hour). The asterisk indicates the beginning of the next light period at LD02. The 514-nm laser (Fianium ultrafast fiber laser SC-400-4-PP) was used to excite mCherry and mCitrine at the same time. The mCherry and mCitrine fluorescence signal were divided by a 593 nm dichroic (Semrock) and filtered by bandpass filters (BP705/100 for mCherry, Chroma; BP510-555 for mCitrine, Edmund Optics). All the fluorescent photons were collected using a high-speed single-photon counting module (Becker & Hickl HPM-100-40).

C) The third major finding is predicted structures of the ROC15-CRY complex. This data is dependent on SAXS and computational efforts. It is therefore subject to the same limitations and concerns regarding SAXS data reporting, interpretation and modeling as discussed in A, with one other large concern.

The sequence of ROC15 is dominated by low complexity regions. It is predicted to be >90% intrinsically disordered. The structures presented here report it as not only well folded, but mostly helical. There is no chance that the sequence below is well structured, which indicates substantial issues with methodology used to predict the 3-dimensional structure.

>tr|B1B5J3|B1B5J3_CHLRE RHYTHM OF CHLOROPLAST 15 OS=Chlamydomonas reinhardtii
OX=3055 GN=ROC15 PE=2 SV=1

MAFDASGALQSCYGAKEDANMDPAAGLVPLTSSLIPPLLAHAFALPQLAQAPAPAQLQPQ
ATAQGYGSMSPALMGALGYATGASYAPVQQQQPSHLASLQYGYLHALATASQQQQQQQ
QQQQAALLAHAHNAALAAVRAGQATSSAMSHPLPPAASADTGSARGAGPSASEPER
AASASVATNSSGALRGSSASYGSAAAAAYAHPLMGLYAALSQQQQAAAYGASPANSY
GAMAPFTTHLHPAYYARPQDAGQAAAAAAAAAAGAYGYAQPTASYGSSFGHPASSYGG
APSSILDFSSSLAAAYRPVAMPAMPAASALAAAAVAAAAGAMPQTSQSVGMTGAVAAGG
GANGAVSSLGGGAAGMMGEEYADDGTTTRAVKRPRLVWTPQLHRKFESAVIKLGEDKAVPK
TIMQEMNIDGLTRENVAHLQYRMIKRRDVTGTSSDGGRDSGTTAAPTASAGTAAAAQRQ
AQQQQRPSPDGATAADGTAGCSPAAVSSPAVAAAAPPSTAAAAATPSAPHSAPSTHGQ
GSSGSGSGCGSGSGSGSGHSGSSARAGSKRSEPEPPSRPTQRAVAVTEAALASSAHP
AGSSGSGRNSAGGSAAAATAAAAGNGVAVMA

Reply:

The proposed structural models are cross validated by comparing secondary structures, trRosetta prediction, multiple independent SAXS-driven MD simulations, and *ab initio* low resolution reconstructions. Detailed results from SAXS data analysis are included in the revised manuscript.

Regarding the question on ROC15 interaction with *CraCRY*, we want to clarify that *in vitro* experiments of ROC15 were performed with purified active domain of ROC15, which is called GARP (Glutamic Acid -Rich Protein) motif. GARP motifs of ROC proteins are a subclass of the MYB DNA-binding domain, similar to those of clock protein PHYTOCLOCK1 (PCL1) (LUX ARRHYTHMO [LUX]) and B-type Arabidopsis response regulators (ARRs) of higher plants (Matsuo, T. et al. *Genes Dev* 22, 918–930 (2008)). Crystal structures shows that the GARP motifs contain well-defined α -helix structures (such as PDB structures 2Y44 and 1IRZ), consistent with the predicted structure of ROC15(GARP).

Subsequently, we focused on the interactions between the GARP motif of ROC15 and the *CraCRY*. Among 10 best docked complexes predicted by ZDOCK, GARP motif was found in a pocket close to the FAD binding site (the complex structures are included in the supplementary materials, Fig. S6c). Therefore, we propose that the GARP domain interaction interface is located in this pocket. As pointed out by the reviewer, the full-length ROC15 has flexible loops connecting structured domains, therefore allowing its interactions with *CraCRY* in a broader region centered around the binding site of GARP. The model of GST-GARP/*CraCRY* complex presented in the manuscript is consistent with the SAXS data, revealing a plausible structural model of this complex.

Minor Comments and Questions:

1. The introduction is riddled with errors or poor references. A few examples:
 - a. Line 38: The authors indicate that mammalian CRYs entrain the circadian clock, that isn't an accurate statement. They act as the primary repressor of the central oscillator.

Reply:

We thank the reviewer for the comment.

Line 38-39 (updated Line 38-41) were changed to 'CRYs also entrain the circadian rhythm in insects and act as the primary repressor of the central oscillator in mammals²⁻⁴. Besides, CRYs have been found to participate in the magnetic navigation of insects and migratory birds⁵⁻⁸.'

We added a new reference about cryptochrome 4 by Xu et al. (reference #8).

- b. Referencing S588D in mammalian CRYs as evidence of CTE regulation is not the best choice. Better choices include the recent discovery that exon11 in the CTE dictates familial advanced phase sleep syndrome (Young lab), competitive interactions between PER and the CTE gate circadian regulation (Partch lab), and/or that the CTE dictates isoform selectivity of Clock drugs (Hirota lab).

Reply:

The manuscript is revised to ‘The exon 11 in the CTE of humanCRY1 deletion mutant lengthens circadian period to cause delayed sleep phase disorder (DSPD), suggesting that the CTE is necessary and sufficient to controls circadian timing by regulating its association with CLOCK:BMAL1¹³.’ (updated Line 51-54)

The work by Partch lab is also included in the updated reference (#13, Parico, G. C. G. et al. The human CRY1 tail controls circadian timing by regulating its association with CLOCK:BMAL1. *Proc Natl Acad Sci U S A.* 117, 27971–27979 (2020)).

- c. Line 51-53: The 464-497 helix in CICY4 is not part of the CTE, it is part of the PHR domain. Further, the CRY4-MagR complex has been refuted by multiple reports. A better reference to the CICY4 CTE would be the determination of its structure and CTE conformational changes by the Takahashii lab (PNAS 2019).

Reply:

Line 51-53 (updated Line 54-56) are changed to “The presence of CTE affects the quantum yield (QY) of radical pair and the order-disordered transition of the phosphate binding motif in *Columba livia* CRY4 PHR domain (PBM motif, 235-245 aa) ^{Error! Reference source not found.}”

The work by Takahashii lab is included in the reference (#14, Zoltowski, B. D. et al. Chemical and structural analysis of a photoactive vertebrate cryptochrome from pigeon. *PNAS* 116, 19449–19457 (2019)).

- d. Line 67-69. The statement implying no direct evidence has been obtained of the conformational changes in CTE and the interactions of these CTEs with the PHR domains is untrue. As they note, the structure of FL DmCRY has been obtained. Time resolved SAXS has been acquired for DmCRY verifying C-terminal conformational changes directly. The Partch lab has employed both structural and direct biophysical techniques to examine PHR-CTE interactions. Oliver-Essen has examined CraCRY conformational dynamics using HDX-MS. And there are numerous others.

It is true there are few atomic resolution structures of intact CTEs (DmCRY is the only one), and that there is limited characterization of Plant CRYs. But, that is not what the authors state.

Reply:

Here, by ‘direct evidence’, we mean measuring the conformation in terms of a spatial parameter, instead of degree of exposure, such as information inferred from the Trypsin

digestion or H-D exchange experiments. To avoid confusion, we changed in the main text updated Line 69-71: ‘However, no direct evidence in terms of a spatial parameter, such as distance between residues, has been reported on the conformational changes in CTEs and the interactions of these CTEs with PHR domains.’

Thanks for pointing out the relevant experiments. We have cited the above references and explained the novelty of the present work in the revised manuscript.

2. Why were the following PDBs chosen to reconstruct a model, they are mostly photolyases and don’t include the actual structure of *CraCRY* PHR.

3FY4	Arabidopsis 6-4 photolyase
2WB2	Drosophila 6-4 photolyase
4I6J	mCRY in complex with SCF complex
IOWL	Photolyase

Were the accessory domains (e.g. SCF complex, bound DNA) removed prior to structure building? What about removal of any CTEs?

Reply:

These templates were selected by the computer programs automatically based on designed algorithm that performs well in CASP contests (Joo et al., Data-Assisted Protein Structure Modeling by Global Optimization in CASP12. *Proteins: Structure, Function, and Bioinformatics*, 86 (S1), 240-246 (2018)).

In fact, *CraCRY* protein resembles photolyase. The PHR domain sequences of cryptochrome and photolyase are highly homologous and conserved. Functionally, both photolyase and *CraCRY* can repair DNA damages. There are more photolyase structures determined than cryptochromes, providing diverse templates that help model building. Sequence evolution analysis reveals that *CraCRY* is an animal-like ortholog, although *CraCRY* is from *Chlamydomonas reinhardtii* algae. Using a diverse set of structure templates from different species often improve model quality. The template selection criteria are summarized in the data-assisted protein structure modeling algorithm. Although the crystal structure of PHR domain (PDB: 5ZM0) was not used as a template, the predicted structure is highly similar to 5ZM0 in its PHR domain (backbone RMSD<1.0 Å). The full-length *CraCRY* structure building is based on both constraints of structural templates and dark-state SAXS data. The sequence alignment was the criterion to determine the structure regions to be used as templates. Prior to structure building, the accessory domains or ligands are excluded.

We clarified this issue in the revision (Method section, updated in Page 17):

“Prior to structure building, the accessory domains or ligands are excluded. Protein complex structure docking was done using ZDOCK 3.0.2 version with default parameters^{Error! Reference source not found.}”

3. Why is the data fit in Supplemental Figure 4a so poor?

Reply:

We obtained an improved fitting to *CraCRY* dimer SAXS data using the dark-state *CraCRY* structure (according to smFRET data) and presented the results in the revised manuscript. We first took predicted structure as the subunits to construct a dimer model, and found the fitting to SAXS data of dimeric *CraCRY* is not good. Then we tried with the refined model obtained from MD

simulations, and obtained the fitting results presented in the current version (revised Fig. S6). The improved data-model consistency implies that the SAXS-driven MD simulations have their merit in structure refinement. However, the signal-to-noise ratios are not high for this dataset due to low sample concentrations of *CraCRY* dimer after the filtration of SEC. The noise levels for q -values beyond $0.15/\text{\AA}$ are larger than the SAXS profiles of monomeric *CraCRY* that went through the same SEC-SAXS measurement. New protocols are needed to overcome this inherent limitation of SEC-SAXS experiments.

- Why are there so many bands in the GST pull down assays (Loaded), the samples look like they are suffering significant proteolysis and are extremely heterogeneous. Are these what the samples look like before the single molecule FRET, if so the sample variability will limit the accuracy of results.

Reply:

In GST pull-down assays, in order to avoid loss of protein activity caused by excessive manipulation, both the *CraCRY*(wt) and FRET construct are only subjected to His affinity column, without further processed with ion exchange columns or Superdex200 (GE). Because of the high specificity of ROC15(GARP)-*CraCRY* interactions and clear light-dependence in bands, impurities will not affect the accuracy of results in pull-down assays. The specificity of this light-dependent interaction is clearly recognizable from the bands.

Unlike the GST pull-down experiments, before single molecule FRET assays, the FRET construct (protein) were purified by both His affinity column and Mono Q column. After incubation with FRET pair dyes, Superdex200 (GE) was used to remove the free dyes. Only the samples around the peak were collected and used in single molecule FRET assay. (The purity of samples is shown in Fig. R9, and Fig. S10 in supplementary materials.)

Fig. R9 (Fig. S10). SDS-PAGE images of *CraCRY*(wt) and FRET-construct during protein expression and purification process. a. Expression of *CraCRY*(wt) without (-) /with (+) IPTG induction. The green arrows indicate where the protein should appear. **b.** *CraCRY*(wt) purified through Ni affinity chromatography. **c.** *CraCRY*(wt) purified through SP column by gradient elution. **d.** Expression of FRET-construct without (-)

/with (+) IPTG induction. The green arrows indicate where the protein should appear. **e.** FRET-construct purified through Ni affinity chromatography. The detergent in lysis buffer was changed to 0.2% CHAPS compared with *CraCRY*(wt). **f.** FRET-construct purified through Mono Q column by gradient elution. The elution bands of Mono Q column from column 7 to 9 are purer than 100% NiB elution of Ni affinity column.

5. Franz-Badur et al. demonstrated the need to use DNase 1 and a heparin column to remove nucleotides that can bind to *CraCRY*. They found it important for sample homogeneity. Why were those procedures not used here?

Reply:

We purified *CraCRY*(wt) by heparin column with DNase as described in Franz-Badur et al. and tested the blue-light induced dimerization of *CraCRY*(wt) by SEC assays. No significant difference was found between the SEC results (**Fig. R10**) and *CraCRY*(wt) purified by SP column as before (**Fig. S3**). SP column and heparin column have same purification procedure to remove nucleotides and separate full-length *CraCRY* from degraded protein.

Fig. R10. The *CraCRY* protein purification results. a. *CraCRY*(wt) purified through Ni affinity chromatography and heparin column by gradient elution. The method is the same as the one described in Franz-Badur et al. , and the green arrow indicates where the protein should appear. **b.** SEC curves of *CraCRY* under dark and lit conditions. The dotted line is the calibration curve from the mixture of monomeric and dimeric BSA proteins, of which the molecular weight is about 66 kDa, very close to *CraCRY*(wt) (70 kDa).

6. For the FRET experiments, a C482A mutant was used. It is believed see Oldemeyer et al., that C482 might play a role in regulating photochemistry through quenching of the tyrosyl radical. Did the authors verify that the C482A mutant retained WT-like photochemistry?

Reply:

We checked the photochemistry of C482A mutant. C482A mutant showed similar results in both blue-light induced spectra changes of FAD (Figure R11a) and the blue-light induced dimerization (Figure R11b), comparing to the wild type *CraCRY*. These results are different from Oldemeyer et al. We speculate that the reducing reagent such as DTT or β -mercaptoethano (BME) used during protein expression and purification are the source of the different results. These reducing reagents will influence the redox states of FAD upon blue-light illumination, then further

affect the dimerization behaviors of *CraCRY* (Figure R1). In our experiments, we avoided the usage of such reducing agents and observed light-induced responses and dimerization of *CraCRY*.

Fig. R11. Photoreduction and dimerization of C482A mutant. **a.** Photoreduction of C482A. These absorption spectra were monitored by UV-VIS spectrometer. Prior to blue-light illumination, C482A *CraCRY* is in FAD oxidized state (black line, “C482A Dark”). After 10-120 seconds of blue-light illumination, the C482A is partially reduced to FADH state (red line, “C482A BL 10 s”; green line, “C482A BL 120 s”). This process is reversible, the reduced protein returns to oxidized state after 12-min relaxation in dark condition. **b.** SEC curves of C482A under dark and lit conditions. The dotted line is the calibration curve from the mixture of monomeric and dimeric BSA proteins, of which the molecular weight is about 66 kDa, close to *CraCRY*(wt) (70 kDa).

- Line 248-249: The authors claim Oldemeyer (misspelled in the manuscript), did not conduct control experiments in the absence of red-light. This is not true and is in Figure 5A of the referenced paper.

Reply:

We apologize for the spelling mistake. The ‘red-light’ should be ‘blue-light’ (corrected in the revision). The light-response process of *CraCRY* we focused on is different from Oldemeyer et al. Oldemeyer et al. studied the process upon **red-light** illumination with blue-light pre-illumination, while we focused on the **blue-light** induced behaviors of *CraCRY* starting from dark condition/state without pre-illumination.

The author name is also corrected in the revision. The corresponding sentence (updated in page 8) was changed to ‘However, **Oldemeyer** et al. found that red-light illumination (at 633 nm) promoted the transform of dimeric *CraCRY*s into oligomers; however, control experiments testing the blue-light induced behaviors of *CraCRY* starting from dark condition/state were lacking⁴⁷.’

- Line 265-266. There data indicates that the Cryo_EM structure of a different protein is consistent with their SAXS data. It does not confirm any dimer interface, as the authors claim. The authors didn’t try any other dimer structures, nor did they prepare/provide any envelope calculations from the raw SAXS data.

Reply:

The SAXS data was not compared to the CryoEM structure of a plant CRY directly. We constructed the dimer structures for *CraCRY* based on the CryoEM model of this CRY protein.

Then we compared the SAXS data with the constructed dimer model. We agree with the reviewer that the consistency with SAXS data is necessary for a correct model, yet not sufficient to identify detailed interface of complexes. Besides the dimer structure built based on the Cryo-EM model, we constructed two other dimer models based on crystal packing. It turned out that the CryoEM model is the most plausible dimer model. In our revised manuscript, the envelopes of SAXS data are also included to support the model based on the Cryo-EM structure template (see Fig. S6a).

Per the suggestion, we rephrased the corresponding sentences (updated in Page 9):

“Besides the proposed dimer model based on the Cryo-EM structure, we also generated alternative dimer models based on crystal packing or biological assembly structures (see Supplementary Fig. S6a). After comparing the goodness-of-fitting to the SAXS data and the low-resolution model, we found that the PHR dimer model based on the Cryo-EM structure is the most plausible option.”

9. In general The methodology and supporting information on most of their approaches is lacking, and renders it near impossible to analyze the validity of their results, and completely irreproducible. As an example: “using the molecular docking methods, the structures ...”. The method referenced has a lot of variables that can impact its validity and results. Specifics for these approaches need to be provided? The information for most computational, SAXS, and structural methods are rudimentary at best.

Reply:

We tried our best to provide sufficient experimental details, so that other labs can repeat the study. Specifically, the smFRET experiments should be readily reproducible by following the information described in the manuscript, see the Method section for experimental protocols, protein sequence information, protein expression/purification/characterization, and constructs for fluorescence experiments.

Thanks to the reviewer for the comment, we added detailed information on computational modeling/simulation and the analysis of SEC-SAXS data that should enhance the presentation. In particular, this revision is supplemented by detailed SAXS data analysis that was very brief in the original submission. We included all basic analysis on SAXS data, including Guinier analysis, dimensionless Kratky plot analysis, $p(r)$ analysis, low resolution model construction etc. For the computational modeling, such as structure prediction, molecular docking, MD simulations, we followed standard protocols and used default parameters unless specified. In particular, (1) the structure prediction is done with two widely used algorithms, the choice of template structures was made by the program, without human intervention; (2) computational docking was also executed on the Zdock webserver, with default parameters; (3) SAXS-driven MD simulations were carried out by following the instructions from Hub's group. These updates are mainly described in the revised Method section. Thanks for pointing out this issue, we made clarification and hope that the revised version contains all information needed for other researchers to validate the results.

10. Line 339-340: The authors claim that “only indirect evidence has been obtained on CTE conformational changes to substantiate the speculation” while this is true since no structures exist of both a light- and dark-state CRY with intact CTE, their claim that they have made the first direct measurement is false. Their method is also indirect, as it relies

on FRET pairs as a reporter of light-induced conformational changes. Time-resolved SAXS studies of *DmCRY*, limited proteolysis studies of numerous CRYs, as well as both FTIR (Kottke lab) and HDX-MS (Oliver-Essen lab) of *CraCRY* are as direct as those reported here.

What is new, and intriguing, is that these are the first single-molecule studies of CRY-CTE dynamics. This is an important take home message and is a meaningful addition to the field, but making larger/broader and inaccurate claims are hurting the take home message.

Reply:

In this study, we report the light-induced conformational changes of *CraCRY* using smFRET method, which measures the distance between labelled sites (C317 of PHR domain and E595C of the CTE). Therefore, as described in the article, this directly distinguished two states of *CraCRY* in terms of a structure parameter. We agree with the reviewer that overclaims should be avoided, since we did not resolve high resolution structures of *CraCRY* in dark/lit conditions. In both the original and revised manuscript, we claimed that the light-induced conformational changes revealed by smFRET are the key discovery. In order to further pursue the structural basis, we proposed, rather than determined, the molecular models based on information from smFRET, homology modeling, and SAXS-MD refinement. These proposed models are subjected to validation from further experiments. Nonetheless, we think that the results are valuable to the research community of CRY proteins. Previous studies, including time-resolved SAXS studies of *DmCRY*, proteolysis studies of CRYs, FTIR (Kottke lab) and HDX-MS (Oliver-Essen lab) of *CraCRY*, did not reveal conformational changes that are quantified as structural parameters. The changes of conformation in these experiments were inferred from the overall structural changes or a potentially more exposed hydrogen which has been replaced by deuterium in bulk system. Compared with these methods, experiments in single-molecule level can give more detailed and direct structural information. We added a note in the discussion (Page 11):

“The proposed structural models are based on information gathered from smFRET experiments, homology modeling, SAXS-driven MD simulations. These models serve as initial structures that can be validated or improved by further studies, such as high-resolution structure determination experiments.”

Reviewer #2:

The manuscript “Direct Experimental Observation of Blue-Light-Induced Conformational Change and Intermolecular Interactions of Cryptochrome” is a **powerful solution-state study leveraging the strengths of FRET, SAXS and advanced MD modelling to understand the structure and mechanism of change undergone by light sensitive receptor proteins**. The manuscript provides structural insights on *CraCRY*, the *C. reinhardtii* ortholog involved in blue-light mediated gene transcription and DNA repair. The authors present in vitro pull-down and in vivo bimolecular fluorescent complementation assays to identify direct binding partners of *CraCRY* and also characterize dimerization state of *CraCRY* using solution-state modelling and analytical size-exclusion chromatography. **The authors show a strong role of the PHR domain that it is involved in mediating interactions with ROC15 and responsible for the observed *CraCRY* dimerization**. I appreciate the correspondence between the FRET studies revealing ~15 Angstrom change that is supported by the SAXS-MD modeling. I recommend the manuscript be revised.

Reply:

Thanks for the constructive comments.

Besides identifying the roles of PHR domain in dimerization and complex formation with ROC15, we applied smFRET method to observe *CraCRY* conformational changes induced by blue light illumination. The exclusion relations between *CraCRY* dimerization and ROC15/*CraCRY* interactions were also reported based on the single molecular experiments. Furthermore, by incorporating SEC-SAXS data under both dark and lit conditions, we refined the structures of full-length *CraCRY*, and obtained plausible structural explanation for the smFRET signal changes.

We appreciate the opportunity to revised the manuscript, and the point-to-point responses are listed in the following.

The experiments are thorough, well supported however, some issues need to be address prior to publication. The use of Chen and Hub’s SAXS-directed molecular dynamics approach to refinement is outstanding. This is exactly how structures should be refined when using SAXS data as opposed to the blind conformational space searching that is typically employed when using CRY SOL or FOXS. As Chen and Hub demonstrated in their seminal paper, MD is incapable of transitioning to the correct states without the added experimental data incorporated during the gradient descent. However, these algorithms will only work well if, 1) the SAXS data represents a single conformational state and 2) the SAXS data is free from instrumentation or measurement artefacts. The authors have addressed 1) by performing inline SEC-SAXS to provide optimal background subtraction as well as resolution of oligomeric species. Point 2 is harder to discern but a major contributor to it would be capillary fouling or aggregation during X-ray exposure.

Reply:

The SEC-SAXS indeed worked well in separating the monomeric and dimeric states of proteins, for both full-length *CraCRY* and its PHR domain. Without the inline SEC, we were stuck with the SAXS data interpretation of mixed signals from monomer and dimer molecules. Regarding the instrumentation, the 19U2 beamline at the SSRF is well characterized and optimized for BioSAXS experiments to reduce instrumentation artefacts. Furthermore, for each sample, we conducted control experiments by varying the light conditions (dark vs lit), and the

contrast clearly showed the blue-light induced dynamics and the interactions between molecules. During the SEC-SAXS, the samples were continuously flowing through the capillary, such that the radiation damages were low. For each dataset, we collected 20 scattering patterns around the elution volume peaks, and found no signs of sample aggregation during X-ray exposure.

In the model-independent characterisation of the conformational changes using FRET, the authors were able to show a change from 50 to 65 Angstroms between residue C317 and E595C when exposed to blue light. This can be interpreted as a compaction of the protein with respect to the analytical SEC (Supporting Fig 3a) where dark-state CraCRY elutes at a slight longer time than lit-state. This model independent analysis could be extended further to the SAXS data as the radius-of-gyration should show a notable change between the lit and dark states of monomeric CraCRY as well as d_{max} , and the corresponding $P(r)$ -distributions. An important assessment to make when performing molecular modelling is if the data could/should be represented by a single conformation. The dimensionless Kratky plot is an excellent tool to show how the thermodynamic state of the protein changes between the dark and lit states and I would recommend that a plot using data from Figure 2 be made with the dimensionless Kratky plot and $P(r)$ -distributions normalised to particle volume. Dimensionless Kratky requires R_g and I_{Zero} to be determined and if the lit-state shows a peak shift towards the upper right quadrant, then it likely suggests the protein is becoming more flexible and that the fitting may require more than one conformation. It's hard to discern the quality of the fits in Fig 2, but by visual inspection, the lit-state is not described in low- q by the model, the R_g of the data is much larger than the model. Residual plots for all fits should be shown, the chi-statistic is hiding the poor fit in the low to medium q region.

Reply:

Thanks for the detailed comments.

The details on SAXS analysis were omitted in the original submission. Thanks for the suggestion and opportunity, we added the results from detailed analysis on SAXS data and the derived structural parameters are summarized as a supplementary note in the format recommended by the SAXS community. The structural parameters include $I(0)$, d_{max} , R_g , etc. The $P(r)$ functions for full-length CraCRY are compared between dark and lit conditions (Fig. 2g). The SAXS data in the measured q -range (up to $\sim 0.47/\text{\AA}$) and the dimensionless Kratky plots for all datasets are included in the revised manuscript. According to the dimensionless Kratky plots, the monomeric CraCRY or PHR exhibit well-defined conformations, while the dimeric states are more flexible. As suggested, we also include residual plots for all fitting analyses. The difference between dark and lit state of monomeric CraCRY is not obvious from the dimensionless Kratky plots.

After conducting extensive SAXS-MD simulations, we found that CTE becomes loosely packed to the PHR domain (see supplementary movies). The MD simulation refined structures converged to two structural ensembles with distinct features on how the CTE is packed to the PHR. Using information from smFRET experiments, we assign the structure ensembles to dark and lit states. The difference between dark and lit states of CraCRY is related to the relative orientation of CTE with respect to PHR: there is a lateral movement of about 22 \AA at the C-terminal (Fig. 2f).

The modelling of the dimeric form of CraCRY is not supported by the SAXS data. Again, a residual plot will show this with Figure 4a, the low-chi is an artefact of the fit as it is clear from

the figure, the fit oscillates and the chi-statistic is largely driven by the magnitude of the errors in the fit. The PHR dimer model-data fit is more believable.

Reply:

The dimer *CraCRY* structure is updated using the dark-state structure obtained from MD simulations, and the data range is restricted up to $\sim 0.17/\text{\AA}$ as suggested by GNOM program, to avoid artefact from noisy signals in larger q values. The residual plots are included for dimeric model fitting analysis (Fig. S6 in the revised manuscript).

Again, with respect to the *CraCRY*-GARP complex (Fig. S5), the fit is not accurately described, the chi-statistic is being driven by the noise and in this case, the authors fit the entire q -range - partly to drive down the chi-statistic which is misleading. I would suggest a dimensionless Kratky plot be made of the lit and dark states of monomeric *CraCRY*, GST-GARP+*CraCRY* data in SFig5, and of the data in SFig4. It may be revealing to show that the binding of GST-GARP resembles the monomeric lit state suggesting binding does not change the compacted state. It is clear from the fit, the low- q region is not explained by any part of the model.

Reply:

The residual plots are included for fitting analysis of the *CraCRY*/GST-GARP complex and *CraCRY* dimers. The dimensionless Kratky plots suggest that the dimeric state is flexible, which may result in the low-quality fitting to the experimental data using a single conformation. As found in the SAXS-MD simulations, the CTE is loosely packed to PHR, allowing the movement of CTE relative to PHR. In addition, it is possible that *CraCRY* subunits can move relatively to each other in their dimeric complex. In the study of the Arabidopsis CRY2, Cryo-EM single molecule imaging data suggest the dimer model is flexible (the dimer electron density map could only be refined to 5.6 \AA , while the map for tetramer was refined to 4.2 \AA). Although the *CraCRY* and CRY2 are different proteins, it is likely that the *CraCRY* dimer structure is also flexible. Therefore, we noted on the flexible structure of *CraCRY* dimer in the revised manuscript (updated in page 11):

“According to SAXS data, the dimeric *CraCRY* or PHR domain exhibits high structural flexibility, possibly caused by multiple conformations or unstable complex structures. This is in accordance of the Cryo-EM models of CRY2 proteins, whose dimeric structure more dynamic than its tetramer³⁷.”

Regarding the *CraCRY*/GST-GARP complexes, the models were constructed based on molecular docking results. The constructed complex structures were compared to the SAXS data to assess the model correctness. Because of the loops in GST-GARP protein, we used applied docking method to analyze the complex structure formed by well-structured GARP domain and *CraCRY*. The proposed complex structure is consistent with SAXS data (fitting to 0.25/ \AA range with residual plots, Fig. S6c). We rewrote this section to present the molecular docking results and the *CraCRY*/GARP complex models (updated in Page 10):

“The GARP motif of ROC15 was docked to a location in the vicinity of the FAD binding site, according to the ten docked structures with the best docking scores. The Raptor-X was applied to predict the structure of GST-tagged GARP domain, which was built into the best complex model of *CraCRY*-ROC15(GARP) (Supplementary Fig. S6c). The *CraCRY*-ROC15(GST-GARP) complex structure is consistent with the SAXS data measured in the SEC-SAXS experiments.”

The data was collected to 0.47 inverse Angstroms, yet was truncated to 0.25 in the presentation in Figure 2, extended slight more in SFig4b and then used entirely in SFig5? Some explanation is required in the text. In some cases, it may be that the data isn't fully transformable to real-space due to poor background matching or capillary following and is truncated to support a P(r)-distribution. The danger of using the noise to higher q-values is it artificially reduces the chi-statistic and this is why residuals are a better guide to confirming the quality of the fit. The residuals should be randomly distributed without oscillating features.

Reply:

Thanks for the comment. The full range SAXS data to 0.47 \AA^{-1} are included in the revised manuscript (Fig. 2a). In the data-model fitting analysis and SAXS-driven MD simulations, we focus on the data range with trustworthy signals, recommended by the P(r) analysis (the q-range is included in the supplementary note for SAXS analysis summary). The model profiles fitting to SAXS data was executed in the truncated ranges, and the figures were drawn with residual plots (Fig. 2 and Fig. S6), as reviewer suggested. To explain the variations in the q-range, we added the following to the Method section (updated in Page 17):

“The SAXS data range for model fitting is truncated to proper regions as suggested by the GNOM program of ATSAS software.”

Minor Comments/Questions:

Figure 1 Legend “The molecular weight is 70195 daltons, very close to the theoretical value at 70195.7 daltons” could be shortened to “ESI determined molecular weight is 70195 (theoretical 70195.7) daltons.”

Reply: This is rephrased as suggested (Page 34).

Figure 3A, UV trace for CraCRY Dark appears to go under the baseline or looks truncated whereas in 3b, the traces clearly above the baseline. I would appreciate it if the authors can show the baseline for the UV trace more clearly in the figure.

Reply:

In the revised Supplemental Figure 3, the UV trace is redrawn. We provided both the SEC-only and SEC-SAXS traces (Supplemental Fig. S3), which are superposed with integrated scattering intensities to indicate the separation of dimeric-/ monomeric CraCRY or PHR domain. As suggested, we expanded the range of the axes to better present the baseline.

In Supporting Figure 4b, how many points are negative? A $q \cdot I(q)$ vs q of the fit would be good to show, however, it would appear that the data above 0.2 inverse angstroms carries little information.

Reply:

The reviewer is correct that the signals are limited for q values beyond $0.2/\text{\AA}$. According to the P(r) analysis, the recommended q_{max} was $0.22/\text{\AA}$, so we fit the PHR dimer model to its SAXS data up to $0.22/\text{\AA}$ (revised Fig. S6).

From Materials and Methods, “Low-resolution envelopes of molecules were computed using

decodeSAXS". I did not see any presentation of low-resolution envelopes. Not relevant to the study.

Reply:

The low-resolution envelopes are added in the revised manuscript. These results were omitted in the original submission for clarity. Thanks for the suggestions from both reviewers, we included the low-resolution envelope models, superimposed to the refined structures (See Fig. 2 in the revised manuscript).

Was the FAD molecule included in the MD simulations? It is not apparent in the supporting movie.

Reply:

The FAD molecule was included in the SAXS-driven MD simulations. During the revision, we carried out several more extensive simulations and the structures are updated as described in revised manuscript. The movies were updated with FAD shown in van der Waals representation (three movies are included in the revision). Furthermore, we conducted comparative model refinement using same approach, started with a different structure predicted by trRosetta algorithm. From these simulations, we observed that the refined structures converged to conformations that are highly similar to our prediction in terms of the overall shapes. The converging results suggest that the constraints due to SAXS data is sufficient to drive the structures towards the correct conformations.

Reviewers' comments:

Reviewer #1 (Remarks to the Author):

I want to applaud the authors for their thorough and complete revision. The revision has addressed most of my concerns and is greatly improved. Overall, the paper is compelling, and provides keen new insight into the structural and biological mechanisms of CraCRY. In my opinion, it offers significant advances that would be of general interest. There remain a few issues that should be corrected, and an exciting discovery in the rebuttal that in my opinion should be highlighted and discussed in the manuscript.

1. The rebuttal indicates some misconceptions regarding existing literature, where previous studies found data that directly conflict with the results presented in this manuscript. The authors' approaches are different, and well supported, and provides more insight to the signaling pathway than the existing manuscript indicates. It is important to indicate in the discussion where discrepancies between approaches still remain, and I'd argue for including some of the data only provided to reviewers, as it provides new insight into signaling (see below).

This is most prevalent in comparing FTIR results from Spexard et al (DOI: [dx.doi.org/10.1021/bi401599](https://doi.org/10.1021/bi401599)), where they observed no secondary structural changes following blue-light illumination.

In contrast to what the authors state, they did examine structural changes using FTIR following initial blue-light exposure.

"The blue light reaction of the oxidized flavin is not accompanied by any detectable changes in secondary structure, in agreement of a role in vivo of an unphysiological preactivation."

Their conclusion was that blue light induced formation of the natural semiquinone did not play a physiological role, rather was only needed to form the semiquinone ground state that responds to red light. This is similar to older literature regarding photolyases, where the oxidized state is likely not biologically relevant, and contrasts from other CRYs (e.g. DmCRY) where they are specifically blue-light sensors.

In Oldemayer et al., they observed monomeric dark-state (Oxidized FAD), partial dimerization in the (FADH⁻ state), and monomers in the FADH⁻ state. This is what the authors observe here.

In both instances, it can be argued that the signaling involves interconverting between FADH states and FADH⁻ states, which can be either red-light or blue-light driven, and the FADox state could be an artifact of purification.

In this manuscript the authors show convincing data of a blue-light response (FRET driven conformational change and blue-light driven complex formation), and strongly suggest a direct biological role in protein-complex formation. The data only supplied as a response to the reviewers may be some of their most intriguing data. Namely, they show that ROC15(GARP) binds to CraCRY in the FADH state, but not in the FADox or FADH⁻ states (mislabeled as FAD⁻ in the response). When you combine this with the in vivo data, it indicates that low blue-light favors FADH and complex formation with ROC15, which is disrupted by high blue-light levels or high (blue+red light). This allows for an adaptive response to changing levels of blue light, or different wavelengths of light in the environment, which is an intriguing result.

These discrepancies, from older literature, and the meaning of the new data should be clearly summarized in discussion or conclusions, and I'd advise including the full aspects of those findings including the meaning of the data provided to reviewers.

2. I'll start by saying the FRET experiments are elegant, and are clear in regards to a blue-light induced conformational change in the CTE. These in many regards are sufficient for further analysis. The SAXS data is internally consistent with the FRET studies, which lends support to the data, however the plots in figure 2 are a bit unconvincing.

Examination of the SAXS data indicates very little difference in scattering data, Kratky plots, and Guinier plots. The only major difference is there appears to be some minor aggregation (likely insignificant) in the dark sample (upward deflection near I0). The other difference can be observed in the pairwise distribution function, where it looks like they used two different Dmax, causing the dark state to be prematurely forced to zero. The raw data does not have significant enough differences to guide a structural change like those discussed, as the observed differences in envelopes could simply result from differences in Dmax.

The FRET data provides similar insight, so I'm not sure it impacts the take-home story regarding a conformational change, but it does suggest the particular conformations shown in the reconstructions may not be accurate. This is amplified by the unusual conformation of the CTE (away from the PHR), whereas Oldemayer argued for the entirety of the CTE docking to the PHR as is seen in all other CRYs characterized.

Minor issues:

In general, I would avoid any use of phrases like "for the first time". There is an instance in the abstract. And later on line 152-153. Finally, it is used on line 360-361. There are a couple more instances of this in the manuscript. Claiming "first" does not add anything to the science, and often when one states that, they are incorrect.

Lines 70-72 remain inaccurate. As mentioned previously, there have been numerous studies of CTE dynamics, including distance measurements. Distance measurements were measured using EPR, in conjunction with various mutants, to delineate a signal transduction pathway in DmCRY.

The paper that made that measurement was published in Communications Biology, <https://doi.org/10.1038/s42003-021-01766-2>.

You are better off simply stating structural information of the CraCRY CTE is largely unknown, due to a lack of atomic resolution structures or distance constraints necessary to refine the CTE structure.

On line 323, it states (GARP motif only) is that the case for all other experiments, as this is the only time that is noted. Also, the domain boundaries (e.g. 1-495) for constructs used, including the GARP domain are not provided. That is expected/required for biophysical studies.

Reviewer #2 (Remarks to the Author):

The manuscript represents an integrated approach at atomistic modelling of the solution state of the lit and dark forms of CraCRY FAD-containing protein. A critical part of this manuscript is the SAXS datasets of the lit and dark forms. To establish that a difference was measured between the lit and dark states of monomeric CraCRY by SAXS (Fig 2), is to show the plot of the ratio of the two SAXS curves. If there are two different form factors, then the ratio should show wiggles (or correlations) that are not shared between the two conditions. If the signals are the same, then the ratio of the two curves would be flat through the specified q-range.

The fact that the dimensionless Kratky plot remains the same between the lit and dark states suggests no major change in the thermodynamic state of the protein. It is either flexible in both states or elongated. The data for the dimers (CraCRY and PHR), while weak, shows the peak corresponding to the Guinier-Kratky Point ($\sqrt{3}$, 1.1) suggesting the dimers are compact globular particles. This is different than the CraCRY-GARP complex, where the peak is shifted up suggesting the particle takes on some flexible characteristics. Nevertheless, the compactness of the CraCRY dimer strongly suggests that there is no flexible domain on the subunit as this would still contribute to moving the peak away from the Guinier-Kratky point which implies the monomer is simply elongated. I think the apparent flexibility in the dimensionless Kratky plot of the dimers is an artefact of the low concentration and background subtraction, you don't have a reliable signal at high q . It's fundamentally determined by the coincidence of the peak at Guinier-Kratky point.

The authors state a light-induced dimerisation model for CraCRY that is subtle and appears to mirror the light sensitive pull down assays with the ROC15 protein. The pull-downs are not showing a stoichiometric response but rather a subtle pull down of the CraCRY protein. What is interesting here is that this subtlety in light induced protein interactions is also evident in the analytical SEC assays which show a small (<10%) formation of dimer when exposed to light. Would you expect this effect to be ameliorated in the presence of BSA? What suggests specificity? Specificity in dimerisation might be inferred by the dimer modelling in S6, however, the authors failed to show the fits of the competing models.

On page 9, second paragraph, the authors discuss a model of the dimerised form of PHR domain and full-length CraCRY in figure S6. Is the dimer of CraCRY using the same dimerisation surfaces that fit the dimer data of PHR domain? Please clarify. Also, the fits or χ^2 values for the alternative models should be demonstrated to inform the reader how poor the alternative models explained the SAXS data.

In addition, the authors stated "the derived structural parameters are summarised as a supplementary note in the format recommended by the SAXS community." This is missing from the revised SI document.

The MD SAXS modeling is sensible and used the TIP3 waters and very reliable force fields. The presence of the SAXS data in the refinement is through a small perturbation of the simulation where the structure is mainly held together through the force fields. A possible computational control would be to use an unstructured CTE domain in the monomeric CraCRY simulation to show it is not consistent with the SAXS observations.

I recommend the manuscript be published with minor revisions.

Minor note:

Page 6 Line 191: It is difficult to distinguish strict flexibility from an elongated particle using the Kratky plot unless it is a fully flexible Gaussian polymer. The dimensionless Kratky plot suggests you have a flexible or elongated particle.

In Figure 2g, how were the distributions scaled? The integrated area of each distribution, when normalised should equal the particle volume. This may help to emphasise any natural differences. In addition, it would be prudent to use more contrasting colours in the overlay of the lit and dark states.

Revision to the reviewers -- COMMSBIO-21-0228A

We highly appreciate the supports and feedbacks from reviewers. We revised the manuscript to incorporate the suggestions, with additional analysis and figures. The contents are enriched by new data, which we presented in this responding letter.

In the revised manuscript, major changes are highlighted in blue color.

Reviewer #1 (Remarks to the Author):

I want to applaud the authors for their thorough and complete revision. The revision has addressed most of my concerns and is greatly improved. Overall, the paper is compelling, and provides keen new insight into the structural and biological mechanisms of *CraCRY*. In my opinion, it offers significant advances that would be of general interest. There remain a few issues that should be corrected, and an exciting discovery in the rebuttal that in my opinion should be highlighted and discussed in the manuscript.

Reply:

We appreciate the reviewer for the careful reading and constructive critics, which really help with the improvement of the presented work. In this round of revision, we took the advices and included more information in the manuscript. The details are explained below.

1. The rebuttal indicates some misconceptions regarding existing literature, where previous studies found data that directly conflict with the results presented in this manuscript. The authors approaches are different, and well supported, and provides more insight to the signaling pathway than the existing manuscript indicates. It is important to indicate in the discussion where discrepancies between approaches still remain, and I'd argue for including some of the data only provided to reviewers, as it provides new insight into signaling (see below).

This is most prevalent in comparing FTIR results from Spexard et al (DOI: [dx.doi.org/10.1021/bi401599z](https://doi.org/10.1021/bi401599z)), where they observed no secondary structural changes following blue-light illumination.

In contrast to what the authors state, they did examine structural changes using FTIR following initial blue-light exposure.

"The blue light reaction of the oxidized flavin is not accompanied by any detectable changes in secondary structure, in agreement of a role in vivo of an unphysiological preactivation."

Their conclusion was that blue light induced formation of the natural semiquinone did not play a physiological role, rather was only needed to form the semiquinone ground state that responds to red light. This is similar to older literature regarding photolyases, where the oxidized state is likely not biologically relevant, and contrasts from other CRYs (e.g. DmCRY) where they are specifically blue-light sensors.

In Oldemayer et al., they observed monomeric dark-state (Oxidized FAD), partial dimerization in the (FADH state), and monomers in the FADH⁻ state. This is what the authors observe here.

In both instances, it can be argued that the signaling involves interconverting between FADH states and FADH⁻ states, which can be either red-light or blue-light driven, and the FADox state could be an artifact of purification.

In this manuscript the authors show convincing data of a blue-light response (FRET driven conformational change and blue-light driven complex formation), and strongly suggest a direct biological role in protein-complex formation. The data only supplied as a response to the reviewers may be some of their most intriguing data. Namely, they show that ROC15(GARP) binds to *CraCRY* in the FADH state, but not in the FADox or FADH⁻ states (misabeled as FAD⁻ in the response). When you combine this with the in vivo data, it indicates that low blue-light favors FADH and complex formation with ROC15, which is disrupted by high blue-light levels or high (blue+red light). This allows for an adaptive response to changing levels of blue light, or different wavelengths of light in the environment, which is an intriguing result.

These discrepancies, from older literature, and the meaning of the new data should be clearly summarized in discussion or conclusions, and I'd advise including the full aspects of those findings including the meaning of the data provided to reviewers.

Reply:

We thank the reviewer for the very profound observation. We have modified our discussion and added two paragraphs to resolve the discrepancies in different experimental methods and reinstate the importance of blue-light to *CraCRY*.

We added the comparison of two other models that describes light-induced conformation changes to the discussion section:

“The conformation and role of CTE has been under intensive investigation. Oldemayer et al. found dimer states via the CTE in dark, and red-light illumination triggers oligomerization. This is in contrast to either Frank-Badur et al. or our work, where *CraCRY*s exist as monomers in dark condition. Using Infrared difference spectroscopy based on the absorption of FAD, Spexard et al. did not register any blue light dependent conformational change in the CTE domain, which is not contiguous to the cofactor FAD. Frank-Badur et al., on the other hand, proposed that the structured CTE binds to PHR in dark condition, but it becomes disordered upon light illumination and dissociates from the PHR domain. Here, based on smFRET and solution X-ray scattering experiments, we provide an alternative model, in which the CTE is partially packed to PHR, and the packing pose is influenced by light. It is likely that the FAD oxidation state affects the CTE packing via the regulation of $\alpha 22$ (the helix adjacent to PHR domain), which is in the vicinity of FAD binding site. This light sensing mechanism is consistent with the model proposed by Frank-Badur et al., although the exact movement of CTE is different.”

Furthermore, the role of red or blue light has been discussed:

“Previous *in vivo* transcription level study of *CraCRY* has shifted the whole focus of this protein to red light responses (DOI: 10.1105/tpc.112.098947). However, our work on the blue-light induced functional dynamics of *CraCRY* has confirmed conformational change, dimerization, and intramolecular interaction can be initiated with UV to blue light. Especially, ROC15 interacts with *CraCRY* when the cofactor is in the FADH• state, which is the red-light absorbing state. The neutral radical state has a relatively flat absorption spectrum covering the whole visible light from 400 to 700 nm (Supplementary Fig. S1b). Therefore, excessive blue light exposure can also promote the transition of FADH• to the fully reduced FADH- state (Supplementary **Fig. R1d**), which is also found in C1CRY4 (DOI: 10.1007/s00018-018-2920-y). This may have a profound implication for *C. reinhardtii* where changing levels of blue light, or different wavelengths of light in the environment can result in an adaptive response.”

Figure R1 (Fig. S12 in the revision). Blue light dependent dimerization, photoreduction and interaction of *CraCRY* with TCEP or DTT. **a.** SEC (green solid) and scattering intensity (orange-yellow solid) curves of *CraCRY* upon blue-light illumination with 10 mM TCEP. While, the dotted lines are SEC (cyan) and scattering intensity (brown) curves of *CraCRY* upon blue-light illumination without TCEP. **b.** SEC (green solid) and scattering intensity (orange-yellow solid) curves of PHR domain under dark and lit conditions with 10 mM TCEP. While, the dotted lines are SEC (cyan) and scattering intensity (brown) curves of PHR domain blue-light illumination without TCEP. **c.** SEC (green solid) curve of *CraCRY* blue-light illumination with 50 mM DTT. While, the cyan dotted line is without DTT. **d.** Photoreduction of purified *CraCRY* with 10 mM DTT. These absorption spectrums were monitored by UV-VIS spectrometer. Before blue-light illumination, the *CraCRY* proteins stay at FAD oxidized state (black line, Dark with DTT). With 10s blue-light illumination, the protein will be partially reduced to FADH[•] state (red line, BL 10s with DTT). And then, *CraCRY* is further reduced to FADH⁻ state after 120 s blue-light illumination (cyan line, BL 120s with DTT). This process is partly reversible. Part of the

reduced protein will return to oxidized state after remaining under dark conditions for 20 min (brown). **e-f.** Pull-down molecular complex analyzed using SDS-PAGE. Green arrows mark the *CraCRY*(wt). Pink arrows mark the band of GST-ROC15(GARP). The images show that *CraCRY*(wt) can not be eluted from the GST column under both dark and lit conditions in the presence of DTT/TCEP. It is worth noting, since the blue-light response in absorption spectrum of *CraCRY*(wt) also changed within DTT/TCEP. The negative interaction results here due to the different spectral response of FAD cannot be ruled out.

2. I'll start by saying the FRET experiments are elegant, and are clear in regards to a blue-light induced conformational change in the CTE. These in many regards are sufficient for further analysis. The SAXS data is internally consistent with the FRET studies, which lends support to the data, however the plots in Figure 2 are a bit unconvincing.

Examination of the SAXS data indicates very little difference in scattering data, Kratky plots, and Guinier plots. The only major differences is there appears to be some minor aggregation (likely insignificant) in the dark sample (upward deflection near I_0). The other difference can be observed in the pairwise distribution function, where it looks like they used two different D_{max} , causing the dark state to be prematurely forced to zero. The raw data does not have significant enough differences to guide a structural change like those discussed, as the observed differences in envelopes could simply result from differences in D_{max} .

The FRET data provides similar insight, so I'm not sure it impacts the take-home story regarding a conformational change, but it does suggest the particular conformations shown in the reconstructions may not be accurate. This is amplified by the unusual conformation of the CTE (away from the PHR), whereas Oldemayer argued for the entirety of the CTE docking to the PHR as is seen in all other CRYs characterized.

Reply:

The proposed models are well founded by the SAXS data. To examine this, we added more analysis to the SAXS data. First, the SAXS profiles for monomeric *CraCRY* in dark/lit states were compared by computing the ratios at each q value, and the difference is insignificant in the region with $q < 0.1/\text{\AA}$. At $q > 0.15/\text{\AA}$, we observed some oscillations around the flat baseline, indicating that the light illumination affects the SAXS profiles (see the new supporting **Fig. R2a**). Secondly, we conduct $p(r)$ profile fitting by varying the D_{max} values. In the previous version, the different D_{max} values were suggested by the Gnom program. For the purpose of cross-validation, we computed the $p(r)$ profiles using the same D_{max} values, and the results are shown in the supporting **Fig. R2b**. Interestingly, we observed the small

broadening of $p(r)$ in the lit state, compared with the dark state $p(r)$, which is similar to the results presented in the main text (**Fig. 2g** and **Fig. R2b**). Furthermore, we computed the $p(r)$ profiles from the PDB models, refined using SAXS-driven MD simulations. Although the details are different from the $p(r)$ derived from SAXS data (possibly due to the solvent layer), we observed that the overall shapes of $p(r)$ calculated from the atomic models in both dark and lit states resemble the SAXS-derived $p(r)$ (see **Fig. R2c**). Therefore, we conclude that the *CraCRY* models are in accordance with smFRET and SAXS experimental data. The structural difference is reflected in the small but meaningful difference in SAXS profiles for dark/lit *CraCRY*. We also carried out conventional MD simulations without applying SAXS data, and found that the CTE converges to a completely different position and is bound to PHR (see **Supplementary Movie 4**). Coincidentally, this position is similar to a Rosetta predicted model, which was subject to SAXS-driven MD simulation (see **Supplementary Movie 3**). Interestingly, the SAXS-driven MD simulation refine the *CraCRY* atomic model towards the structures presented in Figure 2. These two control simulations strongly support the importance of SAXS data in determining the structure of *CraCRY*. As discussed in the article, due to the small difference in SAXS data, the smFRET data were used to distinguish the dark from lit states.

Figure R2 (Fig. S11 in the revision). The comparison of monomeric *CraCRY* SAXS profiles under dark and lit conditions. **a.** The ratio between lit and dark SAXS profiles. The gray color shows the ratio at each q value, and the purple lines indicate the running average with a window size of 20 data points. **b.** The distance distribution $p(r)$ calculated from the dark and lit SAXS profiles, using the same D_{max} (130 \AA). The $p(r)$ of lit state (blue) shows slight expansion compared to the dark state (black), mainly manifested in the regions displayed as insets. **c.** The $p(r)$ calculated from atomic models obtained from SAXS-driven MD simulations. A similar expansion trend is recovered in the atomic models. Note that the differences between (c) and (b) are due to the solvent layer, which is not considered in the atomic models but nonetheless contributes to the SAXS profiles measured in experiments.

The smFRET experiments corroborate that the CTE of *CraCRY* is partially docked to PHR, and the blue light illumination even enhances the dissociation of CTE. This is well-supported by both low-resolution models reconstructed from SAXS data and the atomic models refined using SAXS-driven MD

simulations. We added in-depth discussions on the discrepancy between the present models and the compact models reported in previous studies by Oldemayer et al. or Franz-Badur et al. The comparison of alternative models will allow more experimental testing and validation. We have added a detailed discussion on this issue, please see our response to the first comment.

Minor issues:

In general, I would avoid any use of phrases like “for the first time”. There is an instance in the abstract. And later on line 152-153. Finally, it is used on line 360-361. There are a couple more instances of this in the manuscript. **Claiming “first” does not add anything to the science, and often when one states that, they are incorrect.**

Reply:

We agree with the reviewer that claiming 'first' does not add values to the science. The claims have been removed through the manuscript.

Lines 70-72 remain inaccurate. As mentioned previously, there have been numerous studies of CTE dynamics, including distance measurements. Distance measurements were measured using EPR, in conjunction with various mutants, to delineate a signal transduction pathway in DmCRY.

The paper that made that measurement was published in Communications Biology, <https://doi.org/10.1038/s42003-021-01766-2>.

You are better off simply stating structural information of the *CraCRY* CTE is largely unknown, due to a lack of atomic resolution structures or distance constraints necessary to refine the CTE structure.

Reply:

Thanks for suggesting this interesting and very relevant study. We did not find this work while conducting the literature review. In the revision, we added the work to the introduction and corrected our statement.

On page 2:

"Nonetheless, direct measurements of residue distances are challenging, especially at the single-molecule level. In a recent study, Chandrasekaran et al. applied 4-pulse double electron-electron resonance (4P-DEER) method to investigate light-induced enhanced motion of CTE in DmCRY (Chandrasekaran et al. DOI: <https://doi.org/10.1038/s42003-021-01766-2>). However, the structural information obtained by these bulk methods are insufficient to build atomic resolution structures. And the distance constraints necessary to refine the CTE structure are still lacking for *CraCRY*."

On line 323, it states (GARP motif only) is that the case for all other experiments, as this is the only time that is noted. Also, the domain boundaries (e.g. 1-495) for constructs used, including the GARP domain are not provided. That is expected/required for biophysical studies.

Reply:

We thank the reviewer for pointing out this mistake. We have described the definition of the constructs that we used in the methods and included a figure in the SI. (see **Fig. S14**)

Figure R3 (Fig. S14 in the revision). The constructs of ROC15(GARP) and PHR domain. a. The uniprot ID and amino acid sequence of full-length ROC15. The GARP motif (M377-M445 a.a.) in ROC15 is highlighted in pink color. **b.** The cartoon structure of ROC15(GARP) motif. This structure was predicted by Raptor-X server. **c.** The uniprot ID and amino acid sequence of full-length *CraCRY*. The PHR domain (M1-G495 a.a.) of the *CraCRY* is colored in green. **d.** The cartoon structure of PHR domain, which was downloaded from the protein databank (PDB ID: 5zm0).

Reviewer #2 (Remarks to the Author):

The manuscript represents an integrated approach at atomistic modelling of the solution state of the lit and dark forms of *CraCRY* FAD-containing protein. A critical part of this manuscript is the SAXS datasets of the lit and dark forms. To establish that a difference was measured between the lit and dark states of monomeric *CraCRY* by SAXS (Fig 2), is to show the plot of the ratio of the two SAXS curves. If there are two different form factors, then the ratio should show wiggles (or correlations) that are not shared between the two conditions. If the signals are the same, then the ratio of the two curves would be flat through the specified q-range.

Reply:

Thanks for the suggestion. We computed the ratios between the SAXS intensity profiles at each q-value for monomeric *CraCRY* in its dark and lit states (see Fig. R2 in this responding letter, or Fig. S11 in the revised manuscript). The difference is small, but evident for $q > 0.15/\text{\AA}$, as indicated by the ratio curve. We also re-computed the $p(r)$ distribution functions using the same D_{max} values (see Fig. R2b), and the results are similar to the ones presented in Fig. 2g, indicating that the $p(r)$ analysis results is not due to the D_{max} values optimized by Gnom fitting program. The broadening of the $p(r)$ function in lit state relative to dark state indicates reduced compactness, which is also recovered in the atomic models refined in SAXS-driven MD simulations (see Fig. R2c).

The fact that the dimensionless Kratky plot remains the same between the lit and dark states suggests no major change in the thermodynamic state of the protein. It is either flexible in both states or elongated. The data for the dimers (*CraCRY* and PHR), while weak, shows the peak corresponding to the Guinier-Kratky Point ($\sqrt{3}$, 1.1) suggesting the dimers are compact globular particles. This is different than the *CraCRY*-GARP complex, where the peak is shift up suggesting the particle takes on some flexible characteristics. Nevertheless, the compactness of the *CraCRY* dimer strongly suggests that there is no flexible domain on the subunit as this would still contribute to moving the peak away from the Guinier-Kratky point which implies the monomer is simply elongated. I think the apparent flexibility in the dimensionless Kratky plot of the dimers is an artefact of the low concentration and background subtraction, you don't have a reliable signal at high q. Its fundamentally determined by the coincidence of the peak at Guinier-Kratky point.

Reply:

We agree with the reviewer. The similarity in dimensionless Kratky plots for dark and lit states indicates that similar structural features for monomeric *CraCRY* in both dark and lit conditions (in page 6, “For the full-length *CraCRY*, the peak positions are shifted to the right in the dimensionless Kratky plots,

suggesting flexibility of the CTE or an elongated shape of full-length CraCRY.”). We also checked the SAXS profiles of blue-light induced *CraCRY* dimer and *CraCRY*-ROC15 complexes. We agree with the reviewer that the *CraCRY* dimer is most likely to be in a compact form. In contrast, the *CraCRY*-ROC15 complex data shows an up-lifting trend at larger q-values. However, we cannot rule out the possibility of artefacts due to weaker signals at larger scattering angles, since the SEC-SAXS did not yield high concentrations of dimer or the *CraCRY*-ROC15 complex molecules. We discussed the limitation of the present SAXS experiments and hope to collect stronger signals after improving the experimental protocols. Nonetheless, the blue-light induced *CraCRY* dynamics and interactions with ROC15 are supported by the SEC-SAXS data.

In page 12, we added:

“According to SAXS data (Fig. 2b), the dimeric *CraCRY* or PHR domain are compact, manifested in the peak near the Guinier-Kratky point. There is a slight right-shift of the peak position, indicating either flexible conformations of the dimer or more elongated shape. In the Cryo-EM study of CRY2 proteins, researchers found that dimeric structure is more dynamical than its tetramer (DOI: 10.1038/s41594-020-0420-x). On the other hand, we noticed that the signal-to-noise ratio (SNR) needs to be enhanced by improving the sample quality and SEC-SAXS experimental protocols. The lower SNR at larger scattering angles for dimers or *CraCRY*-ROC15 complexes may contribute to the up-shift observed in Kratky plots.”

The authors state a light-induced dimerisation model for *CraCRY* that is subtle and appears to mirror the light sensitive pull down assays with the ROC15 protein. The pull-downs are not showing a stoichiometric response but rather a subtle pull down of the *CraCRY* protein. What is interesting here is that this subtleness in light induced protein interactions is also evident in the analytical SEC assays which show a small (<10%) formation of dimer when exposed to light. **Would you expect this effect to be ameliorated in the presence of BSA?** What suggests specificity? Specificity in dimerisation might be inferred by the dimer modelling in S6, however, the authors failed to show the fits of the competing models.

Reply:

In the design of our pull-down assay, our *CraCRY* was used as the bait. The blue-light dependent appearance of the *CraCRY* band strongly suggests specificity. In the light of reviewer #2’s comment, we have also conducted the pull-down experiment in the presence of BSA (see **Fig. R4**). The results resemble what has been found in the BSA-free experiment, with no amelioration.

In our *in vitro* single-molecule dimerization assay, the purified *CraCRY* formed homo-dimers. For the specificity of dimerization, please refer to the answer of the next question for the discussion of our chosen

dimer model.

Figure R4. (Fig. S13 in the revision) Blue light dependent dimerization and interactions between ROC15(GARP) and CraCRY with BSA added. a. SEC curves of BSA only (grey dots) and CraCRY with BSA added (blue solid) upon blue-light illumination. P1 and P3 represent the positions of dimeric and monomeric CraCRY (green vertical lines), respectively. While, P2 and P4 represent the positions of dimeric- and monomeric- BSA (Yellow vertical lines), respectively. b. Pull-down molecular complex analyzed using SDS-PAGE. Green arrows mark the CraCRY(wt). Pink arrows mark the band of GST-ROC15(GARP). Yellow arrows mark the BSA. In the loading sample, the mass concentration ratio of CraCRY, GST-GARP and BSA is 1:1:1.2. These images show that both the blue light dependent dimerization and interactions between ROC15(GARP) and CraCRY are specific and will not be altered in the presence of BSA.

On page 9, second paragraph, the authors discuss a model of the dimerised form of PHR domain and full-length CraCRY in figure S6. Is the dimer of CraCRY using the same dimerisation surfaces that fit the dimer data of PHR domain? Please clarify. Also, the fits or chi2 values for the alternative models should be demonstrated to inform the reader how poor the alternative models explained the SAXS data.

Reply:

The light-induced dimerization experiment was carried out with purified CraCRY samples in smFRET, pull-down assays, or SEC-SAXS experiments. The data, especially the control experiments in dark conditions, consistently support that CraCRY dimerization can be triggered by blue-light illumination. Regarding the dimer structures, we constructed 3 models, whose theoretical profiles fitting to experimental data are included in the revision (thanks for the suggestion, see Fig. R5 and Supplementary Fig. S6a in the updated manuscript). As shown in the Fig. R5, the chi-scores are not enough to distinguish the three models, due to the low signal-to-noise ratio in the PHR dimer SAXS data. On the other hand, out of the three models, only the head-to-head model based on the Cryo-EM dimer structure agrees with

the low-resolution envelope model. Based on this information, we selected the PHR dimer structure from the three competing models. Furthermore, in the other two alternative dimer models, the CTE domains caused steric clashes when the full-length *CraCRY* are superposed to the PHR dimer structures. Therefore, we used the same dimerization interface as in the PHR dimer structure model to build the dimer structure for full-length *CraCRY*, and the fitting results indicate very good agreement with the SAXS data.

Besides the updated **Fig. S6**, we included the following in the revision (**Page 9**):

“After superimposing the crystal structure of the PHR domain (PDB code: 5ZM0) on the Cryo-EM model, we obtained a dimer structure model for PHR domain, of which the theoretical SAXS profile is in good agreement with the experimental data (see Supplementary Fig. S6a, the third model, named as the head-to-head model hereafter). Besides the constructed dimer model based on the Cryo-EM structure, we also generated alternative dimer models based on crystal packing or biological assembly structures (see Supplementary **Fig. S6a** for two alternatives). As shown in **Fig. S6a**, all three PHR dimer models are in agreement to SAXS data, yielding to similar chi-scores between theoretical profiles and experimental data. We selected the mostly plausible dimer model based on two additional criteria: (1) the head-to-head model is consistent with the low-resolution envelope model; (2) the head-to-head model do not introduce steric clashes when constructing dimer structure of full-length *CraCRY*, while the other two alternative models result in unphysical clashes at the CTE. Considering that the dimeric *CraCRY* possesses conformations similar to its dark state as revealed in smFRET experiments, the dimer model for full-length *CraCRY* was constructed by superposing the dark state structure to the Cryo-EM structure template. The resulted model is also in good agreement with the SAXS data measured in the SEC-SAXS experiment (see Supplementary **Fig. S6b**).”

Figure R5 (Supplementary Figure S6a). Predicted PHR dimer structure and its theoretical profile fitted to SAXS data. Three models were constructed based on crystal packing and Cryo-EM model templates (the PDB codes are indicated next to the dimer model). The head-to-head dimer model fits to the low-resolution envelope derived from SAXS data (shown as the transparent envelope). The fitting results are shown on the right panels. The residual plot for the chosen model that is based on the Cryo-EM dimer is also shown (right panel, bottom).

In addition, the authors stated “the derived structural parameters are summarised as a supplementary note in the format recommended by the SAXS community.” This is missing from the revised SI document.

Reply:

The supplementary table was uploaded, but mistakenly labelled as Cover Art. We include the table in the supplementary material documents (see **Table S6**).

The MD SAXS modeling is sensible and used the TIP3 waters and very reliable force fields. The presence of the SAXS data in the refinement is through a small perturbation of the simulation where the structure is mainly held together through the force fields. A possible computational control would be to use an unstructured CTE domain in the monomeric *CraCRY* simulation to show it is not consistent with the SAXS observations.

Reply:

It is a good suggestion; however, it is computationally challenging to fold a 100-residue CTE from a

fully unstructured CTE along with the 500-residue PHR domain. Following the suggestion, we carried out the conventional MD simulations for 1000 ns, without applying SAXS profile restraints. In this control simulation, we observed that the CTE tends to bind to PHR, rather than extending from the PHR domain. The contrast between conventional equilibrium MD simulation and SAXS-driven MD simulation clearly indicates the importance of restraints imposed by experimental SAXS data. We added the control simulation as **Movie S4** to supplementary materials to support the necessity to combine experimental SAXS data with simulations.

I recommend the manuscript be published with minor revisions.

Minor note:

Page 6 Line 191: It is difficult to distinguish strict flexibility from an elongated particle using the kratky plot unless it is a fully flexible Gaussian polymer. The dimensionless Kratky plot suggests you have a flexible or elongated particle.

Reply:

We agree with the reviewer and add the “elongated” in the statement.

“For the full-length *CraCRY*, the peak positions are shifted to the right in the dimensionless Kratky plots, suggesting flexibility of the CTE or the elongated shape of full-length *CraCRY*.”

In Figure 2g, how were the distributions scaled? The integrated area of each distribution, when normalised should equal the particle volume. This may help to emphasise any natural differences. In addition, it would be prudent to use more contrasting colours in the overlay of the lit and dark states.

Reply:

The $p(r)$ in **Fig. 2g** were normalized such that the area under the curve is 1.0. Thanks for the suggestion to enhance the contrast in this figure, we showed magnified regions with large differences (see **Fig. R2b**). The overall difference between dark and lit state is small in both the SAXS profiles and the derived $p(r)$ curves. We calculated the $p(r)$ distributions from the refined atomic models and observed a similar trend of broadening in the lit state relative to the dark state.

REVIEWERS' COMMENTS:

Reviewer #1 (Remarks to the Author):

All corrections have been made to my satisfaction. Thank you to the authors for diligently making corrections and additions as requested, the manuscript is greatly improved and represents a significant advance to the field.

Reviewer #3 (Remarks to the Author):

Cryptochromes are photosensory flavoproteins that are known to play key roles in circadian cycle and in magnetic sensing. Using smFRET and SAXS analysis, authors show that carboxy terminal extension (CTE) of *C. reinhardtii* animal-like cryptochrome opens up on irradiation with blue light allowing it to interact with Rhythm Of Chloroplast 15 (ROC15), a circadian-clock-related protein. Blue light induced dimerization of CraCRY was found to be exclusive of its complex formation with ROC.

Authors generate a model of CraCRY dimer based on cryoEM structure of a plant homolog which they refine and validate with the help of SAXS analysis.

The revised manuscript suitably addresses all the concerns raised.

Revision to the reviewers -- COMMSBIO-21-0228C

Reviewer #1 (Remarks to the Author):

All corrections have been made to my satisfaction. Thank you to the authors for diligently making corrections and additions as requested, the manuscript is greatly improved and **represents a significant advance to the field.**

Reviewer #3 (Remarks to the Author):

Cryptochromes are photosensory flavoproteins that are known to play key roles in circadian cycle and in magnetic sensing. Using smFRET and SAXS analysis, authors show that carboxy terminal extension (CTE) of *C. reinhardtii* animal-like cryptochrome opens up on irradiation with blue light allowing it to interact with Rhythm Of Chloroplast 15 (ROC15), a circadian-clock-related protein. Blue light induced dimerization of CraCRY was found to be exclusive of its complex formation with ROC.

Authors generate a model of CraCRY dimer based on cryoEM structure of a plant homolog which they refine and validate with the help of SAXS analysis.

The revised manuscript **suitably addresses all the concerns raised.**

Reply:

Appreciate the time and suggestions from both reviewers. We are glad to hear that the carefully and thoroughly revised manuscript can “suitably address(es) all the concerns raised”, particularly to receive the recognitions as “the manuscript is greatly improved and represents a significant advance to the field”. Furthermore, we hope our work can provide more new ideas and methods for studying more other conformational-change-dependent signaling pathways, not just the molecular functional mechanisms of cryptochromes.